# A comprehensive investigation of intracortical and corticothalamic models of the alpha rhythm

**Sorenza P. Bastiaens**[1,2]*, **Davide Momi**[3,4], **John D. Griffiths**[1,2,5]*

**1** Institute of Medical Sciences, University of Toronto, Toronto, Ontario, Canada, **2** Krembil Centre for Neuroinformatics, Centre for Addiction and Mental Health, Toronto, Ontario, Canada, **3** Department of Psychiatry and Behavioral Sciences, Stanford University Medical Center, Stanford, California, United States of America, **4** Wu Tsai Neurosciences Institute, Stanford University, Stanford, California, United States of America, **5** Department of Psychiatry, University of Toronto, Toronto, Ontario, Canada

* sorenza.bastiaens@gmail.com (SPB); john.griffiths@utoronto.ca (JDG)

## Abstract

The electroencephalographic alpha rhythm is one of the most robustly observed and widely studied empirical phenomena in all of neuroscience. However, despite its extensive implication in a wide range of cognitive processes and clinical pathologies, the mechanisms underlying alpha generation in neural circuits remain poorly understood. In this paper we offer a renewed foundation for research on this question, by undertaking a systematic comparison and synthesis of the most prominent theoretical models of alpha rhythmogenesis in the published literature. We focus on four models, each studied intensively by multiple authors over the past three decades: (i) Jansen-Rit, (ii) Moran-David-Friston, (iii) Robinson-Rennie-Wright, and (iv) Liley-Wright. Several common elements are identified, such as the use of second-order differential equations and sigmoidal potential-to-rate operators to represent population-level neural activity. Major differences are seen in other features such as wiring topologies and conduction delays. Through a series of mathematical analyses and numerical simulations, we nevertheless demonstrate that the selected models can be meaningfully compared, by associating parameters and circuit motifs of analogous biological significance. With this established, we conduct explorations of rate constant and synaptic connectivity parameter spaces, with the aim of identifying common patterns in key behaviours, such as the role of excitatory-inhibitory interactions in the generation of oscillations. Finally, using linear stability analysis we identify two qualitatively different alpha-generating dynamical regimes across the models: (i) noise-driven fluctuations and (ii) self-sustained limit-cycle oscillations, emerging due to an Andronov-Hopf bifurcation. The comprehensive survey and synthesis developed here can, we suggest, be used to help guide future theoretical and experimental work aimed at disambiguating these and other candidate theories of alpha rhythmogenesis.

**Data availability statement:** All related files and programming code are publicly available on GitHub at https://github.com/GriffithsLab/Bastiaens2025_AlphaModels.

**Funding:** This work was supported by the Krembil Foundation (PI: JDG) (https://www.krembilfoundation.ca/), the Labatt Family Network (PI: JDG), the CAMH Discovery Fund (PI: JDG) (https://www.camh.ca/en/science-and-research/discovery-fund), and the Canadian Institutes of Health Research Grant (FRN 183646_1 to PI: B Zrenner, Co-I: JDG). The funders had no role in study design, data collection and analysis, decision to publish, or preparation of the manuscript.

**Competing interests:** The authors have declared that no competing interests exist.

## Author summary

The human brain produces rhythmic patterns of electrical activity that can be measured using electroencephalography (EEG), a non-invasive technique widely used in research and medicine. Among these signals, alpha rhythms are a dominant pattern, linked to relaxation, attention, and neurological disorders. Despite decades of research, the mechanisms behind alpha rhythm generation remain unclear. Our study reviews and compares four leading models of alpha rhythm, each developed by neuroscientists over the last 30 years. These models use mathematical equations to represent how groups of neurons interact to produce oscillations. We identify key similarities, such as their shared use of feedback loops and neural signalling delays, as well as differences in their wiring patterns and assumptions. We also explore how changes in neuronal connections and response rates affect these models' behaviour. Finally, we identify two distinct mechanisms of alpha generation: one based on random fluctuations and another on stable, recurring oscillations. Our work bridges gaps between existing theories and provides a framework to identify the strengths and weaknesses of prominent models in representing alpha rhythms. By clarifying their capabilities and limitations, this study offers valuable tools for advancing future research into the brain's rhythmic activity and its underlying mechanisms.

## Background

### Overview and aims

The classical alpha rhythm is an approximately 8–12 Hz oscillatory activity pattern that is highly prominent in electroencephalogram (EEG), electrocorticogram (ECoG), and local field potential (LFP) recordings from humans and other species, particularly during states of quiet wakefulness (Fig 1A). Almost 100 years after its discovery [1], alpha frequency activity remains one of the most robustly observed and broadly significant phenomena in all of neuroscience, yet also one of the most enigmatic [2]. Alpha plays a fundamental role in a wide range of cognitive processes, and abnormal alpha rhythms are frequently identified in psychiatric and neurological conditions ([3–6]; summarized in Fig 1A, panel 3). However, despite the profound importance of alpha rhythms—both in terms of their undeniable prominence in empirical EEG data, and their implication across a broad range of phenomena across clinical and cognitive neuroscience—their mechanistic physiological basis and functional significance remains unclear. Several theories of alpha rhythmogenesis have been proposed over the years, often emphasizing different physiological substrates such as pacemakers, recurrent activity and excitatory-inhibitory interactions in cortical column microcircuits, or delayed excitatory feedback within cortico-thalamocortical loops (Fig 1B). There have however been relatively few attempts to evaluate and compare in detail these alternative theories in conjunction, and thereby arrive at a useful synthesis of the most compelling accounts. Developing such a synthesis is a principal aim of the present study.

A central criterion around which we base this investigation is the requirement that the models of interest should be expressed in concrete mathematical language, as well as being implemented in numerical simulations and/or quantitative analytic computations. Specifically, we consider a particular class of neurophysiological model—neural population models (NPMs)—that amongst the different mathematical formulations used over the past half century of efforts to model and understand alpha activity (Fig 1C) are the one that has been used most routinely [7–14].

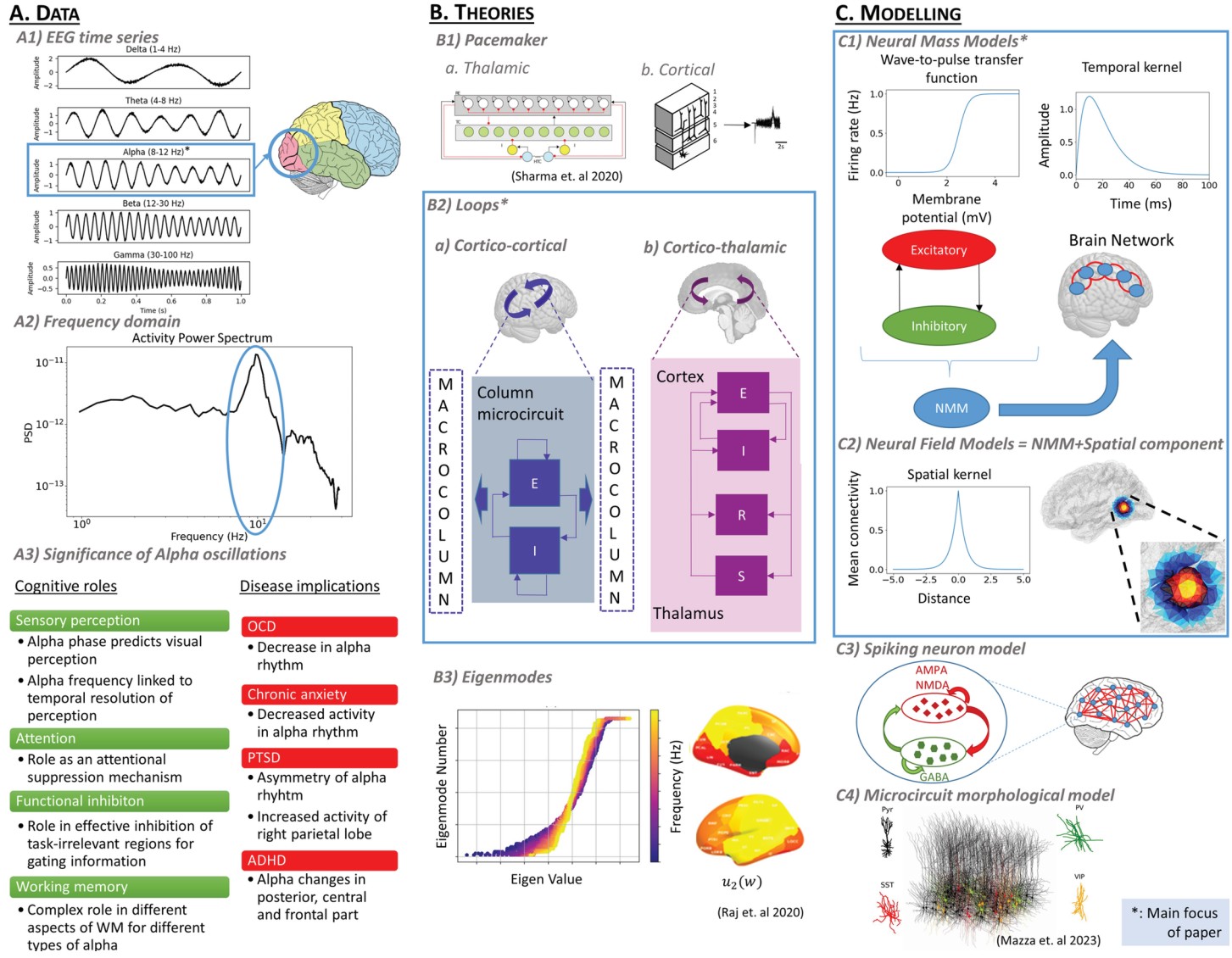

**Fig 1. Data, theories, and models of the EEG alpha rhythm.** (**A**) Alpha oscillations are most strongly observable in the occipital lobe of the cerebral cortex (A1), where they are characterized by a peak in the power spectrum between 8 and 12 Hz (A2). Panel A3 summarizes the role alpha plays in cognitive processes, as well as abnormal alpha rhythm features observed in various diseases (see refs in main text). (**B**) Summary of the different theories that have been proposed to explain the alpha rhythm (thalamic image reference (B1) [18] and eigenmodes reference (B3) [19]). We focus on theories emphasizing the importance of interactions between neural populations (B2). (**C**) Alpha rhythm theories are clarified and concretized by mathematical formulations, allowing numerical and analytical investigation of their predictive and explanatory scope. The principal class of models used to date are neural population (neural mass and neural field) models (C1 and C2), which are the focus of the present work (reference for neurons in C4 [20]).

We focus on four extensively studied NPMs that are commonly used to describe EEG alpha activity in the neuroimaging, neurophysiology, and computational neuroscience literature: the Jansen-Rit (JR; [9]), Moran-David-Friston (MDF; [12,15]), Liley-Wright (LW; [10,16]), and Robinson-Rennie-Wright (RRW; [14,17]) models. These shorthand terms reference certain key individuals who contributed to the conception and/or development of several prominent strands in the research literature. We do note however that they are imperfect ones—both because all of the models studied here build directly on the earlier work of other

important theoreticians (e.g. Freeman, Zetterberg, Lopes Da Silva, Cowan, Nunez), and also in some cases each other (e.g. MDF is an indirect extension of JR). We begin over the next few sections with a description of general elements present in the JR, MDF, LW, and RRW models, and a summary of their individual characteristics. Direct comparisons between each of them are then made, first in the context of the alpha regime, and then extending into other oscillatory regimes at non-alpha frequencies. A central objective in this work is to identify common patterns between the models, using numerical simulations across a broad parameter space to identify the effects of rate constants, inter-population connectivity structure, and other factors on oscillatory dynamics. These similarities and differences across models constitute the points of agreement and divergence across current theories of alpha rhythmogenesis, and it is the mapping of this theoretical landscape that is our main aim in the present paper. The origin, biological significance, and validity of their parameters, as well as the functional forms of their equations, are also considered when discussing the respective limitations and advantages of each candidate model.

## Alpha origins and rhythmogenesis: current theories

Neural oscillations are repetitive, quasiperiodic patterns of brain activity that are believed to play a key role in various sensory-cognitive processes [21]. In humans, oscillations are most commonly captured with EEG, a non-invasive neuroimaging modality that uses scalp-recording electrodes to capture large-scale neuroelectric activity with high temporal resolution. In order to quantify oscillatory activity, the measured signal is typically decomposed into its power spectrum frequency components via a Fourier or related transform, and subsequently often aggregated into several canonical frequency bands. Alpha-frequency oscillations, or 'alpha waves', usually defined as the EEG frequency band between 8 and 12 Hz [22], are associated with quiet wakefulness, meditation, relaxation and reflection [23]. In EEG recordings, alpha waves are most prominent around the occipital lobe when the subject is awake with eyes closed and not engaged in a stimulus-locked cognitive task, also known as the *resting state* [24]. Their role is believed to be fundamental for a number of top-down cognitive processes [23] such as sensory perception [25], attention (as an attentional suppression mechanism—[26]), functional inhibition [6], working memory [27] and long-term memory [28]. Abnormal EEG rhythmic patterns, including aberrant alpha oscillations, are indicative of atypical bioelectrical activity that may suggest the presence of cognitive and/or mental disorders [29–34]. A comprehensive survey of the vast research literature on alpha in cognitive and clinical neuroscience is beyond the scope of the present work; for this we refer the reader to excellent recent treatments by [35] and [36].

Although the alpha rhythm was the first rhythmic wave identified and named by Hans Berger in 1929 [1,37], and is considered the predominant oscillation in the human brain [28] with significant implications in empirical EEG data and various clinical and cognitive neuroscience studies, the physiological mechanisms underlying its generation and functional significance remain poorly understood. Unlike other well-characterized brain oscillations, such as beta and gamma waves, whose neural circuitry is believed to rely on local connectivity alone [38], the generation of alpha rhythm has been proposed to involve contributions from both cortical and thalamic brain regions, which may also influence and interfere with each other [38,39]. Several hypotheses have been proposed regarding the composition and mechanistic organization of these alpha circuits, which can be grouped under three categories: *pacemaker*, *local network*, and *global network* theories. A thorough review of the pacemaker and global network theories are beyond the scope of the present review, but some additional notes and references are provided in Fig 1 and  S10 Appendix, and an extensive discussion can also be

found in [40]. Local network theories propose that alpha rhythms are produced by interactions between excitatory and inhibitory neural populations, typically equipped with dendritic response function terms and saturating nonlinearities [41]. This theory class is by far the most established and extensively studied of the three, and therefore serve as the primary focus of the present paper. Specifically, we examine in detail the two prevailing variants of the local network theory of alpha rhythmogenesis, whose central tenets are the following:

1. Intracortical theory: alpha oscillations are generated by recurrent activity and excitatory-inhibitory interactions within cortical column microcircuits.
2. Corticothalamic theory: alpha oscillations are generated by delayed excitatory feedback within corticothalamocortical loops.

These two accounts (represented in Fig 2) describe the origin of alpha waves as a phenomenon relying on dynamics of local networks of interconnected neural populations, and thus occurring at the *mesoscopic* spatial scale. Computations underlying brain functions such as action, perception, learning, language and higher cognition have been hypothesized to also originate from activity within neural ensembles at this spatial scale [42]. However, whilst current neural recording technologies allow straightforward measurement of macroscale (EEG, MEG, fMRI, ECoG) or microscale (single cell recording, fluorescence calcium imaging, multielectrode arrays) activity, mesoscopic activity is more challenging to measure directly. Although local field potentials (LFPs) provide a proxy for mesoscopic neural activity, they face limitations due to their invasiveness, limited spatial coverage, and the overlapping synaptic currents from diverse neural sources [43]. This complexity, combined with variability in recording and analysis methods, contributes to inconsistent interpretations of LFP-derived oscillatory rhythms. To help bridge the gap between these scales, and explore the rhythmogenic mechanisms entailed by the two local theory types summarized above,

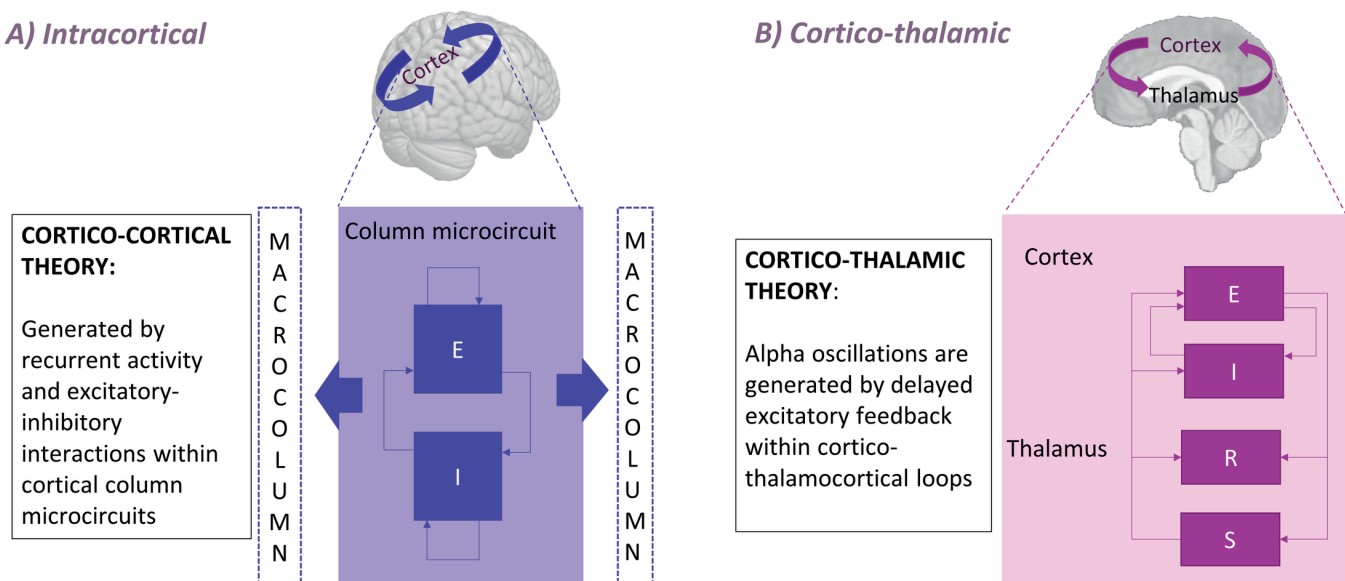

**Fig 2. Schematic depiction of two candidate theories of alpha rhythmogenesis.** (**A**) Cortico-cortical columnar microcircuit model, representing the generation of alpha rhythm through interconnected macrocolumns. (**B**) Cortico-thalamic model, involving thalamic neural populations in the process of alpha genesis.

mathematical models implementing these theories have largely focused on describing and simulating mesoscale neural population wiring and activity.

## Bridging scales: mathematical modelling of mesoscopic neural population dynamics

Mathematical models of human brain activity have provided significant insights into neural processes at multiple scales [42]. The present study focuses on the alpha rhythm using a 'top-down' approach, whereby a mathematical expression is used that represents the collective activity of neuron groups rather than individual cells [44,46]. A general term for this set of techniques is neural ensemble or *neural population models* (NPMs). NPMs consider the aggregate activity of neuron populations with common synaptic connectivity (excitatory or inhibitory), assuming uncorrelated states across the ensemble, and capturing emergent properties of neural tissue patches [47]. This method is effective for modelling oscillatory activity like the alpha rhythm, aligning with the spatial scales of EEG channels and approximating local field potentials [48,49].

NPMs include neural mass models (NMMs), mean-field models (MFMs), and neural field models (NFMs; [42,50])—however, it should be noted that the terminology for these NPM sub-types is not used consistently across the published literature. In one prominent strand of work [42,51] the term 'MFM' is reserved specifically for NPMs that simplify neural population activity using a diffusion approximation, defining it as a standard normal probability distribution characterized by the mean and variance of the firing rate, with stochastic dynamics governed by Fokker-Planck equations. In this schema, NMMs are defined as a specific type of NPM where the variance of the state variables (e.g. average firing rate) across the population is fixed, allowing a simpler representation of the dynamics than MFMs with fewer equations [47]. NMMs are 'point process' models, i.e. they describe neural population activity without any explicit spatial information. In contrast, NFMs include local spatial information by considering the cortex as a smooth and continuously connected sheet, supporting phenomena such as propagating activity waves, often described by damped wave equations [47,52]. Both NMMs and NFMs can be used to simulate whole-brain activity, by coupling local neural populations according to a discrete weighted connectivity matrix ('anatomical connectome'), or a continuous cortical surface manifold, respectively [40,47,53–56]. For a detailed review of NPM development and whole-brain modelling, see [57] and [58], and additional remarks given in S10 Appendix.

## Classification of NPMs and mathematical characteristics of convolution-based models

The three NPM variants discussed above represent three different approaches to the treatment of heterogeneity in mesoscale activity patterns—with NFMs describing variation over space, MFMs describing variation within a point process as a statistical distribution, and NMMs simplifying both of these by assuming no spatial or statistical variation. NPMs can also be categorized according to their general approach to the mathematical description of neural population-level activity, with the main distinctions being convolution vs. conductance-based and voltage vs. activity-based models. Activity-based models, such as the Wilson-Cowan equations [59,60] have state variables representing the proportion of active cells in the population, while voltage-based models represent the population-average membrane potential within various neuron classes. Conductance-based NPMs assume very high coherence between neurons, to the extent that the dynamics of neuron population resembles the dynamics of each single neuron, allowing the use of equations that follow the same structure

as single neuron conductance-based models [47,61], with distinct ionic current types and corresponding channel kinetics. We refer the reader to [51,61–63] for further discussion on the nuances of conductance-based and activity-based modelling approaches, and restrict the remainder of our discussion here to convolutional voltage-based NPM type, of which the four models focused on in this paper are all variants.

Convolution-based NPMs are typically formulated using two types of mathematical operator (Fig 3): the rate-to-potential (or 'pulse-to-wave') operator, describing synaptic and dendritic dynamics, and the potential-to-rate (or 'wave-to-pulse') operator, representing the output firing rate at the soma [44,64–66]. The rate-to-potential operator describes a conversion from an afferent population's firing rate to an excitatory or inhibitory post-synaptic membrane potential, usually in the form of an impulse response. It has been shown that the convolution of the incoming spike rate with an impulse response adequately reproduces the postsynaptic potential in response to presynaptic firing [67,68]. This is expressed as a second-order differential equation, which makes the representation of chemical synapses linear [64,67,69, 70]. The nonlinearity is introduced with the potential-to-rate operator, generally in the form of a sigmoid, which transforms the average membrane potential of the population into the average rate of action potentials fired by the neurons. The sigmoid form is not derived from a biophysical model, but rather seen as a physiologically consistent choice [71], although one that does limit the effective dynamic range [70]—the validity of which we discuss further in Model limitations and critique. In principle, the introduction of nonlinearity through the sigmoid allows these models to capture complex neural dynamics such as chaos. In practice this is rarely of great scientific interest, and indeed the technique of linearizing NPMs around their

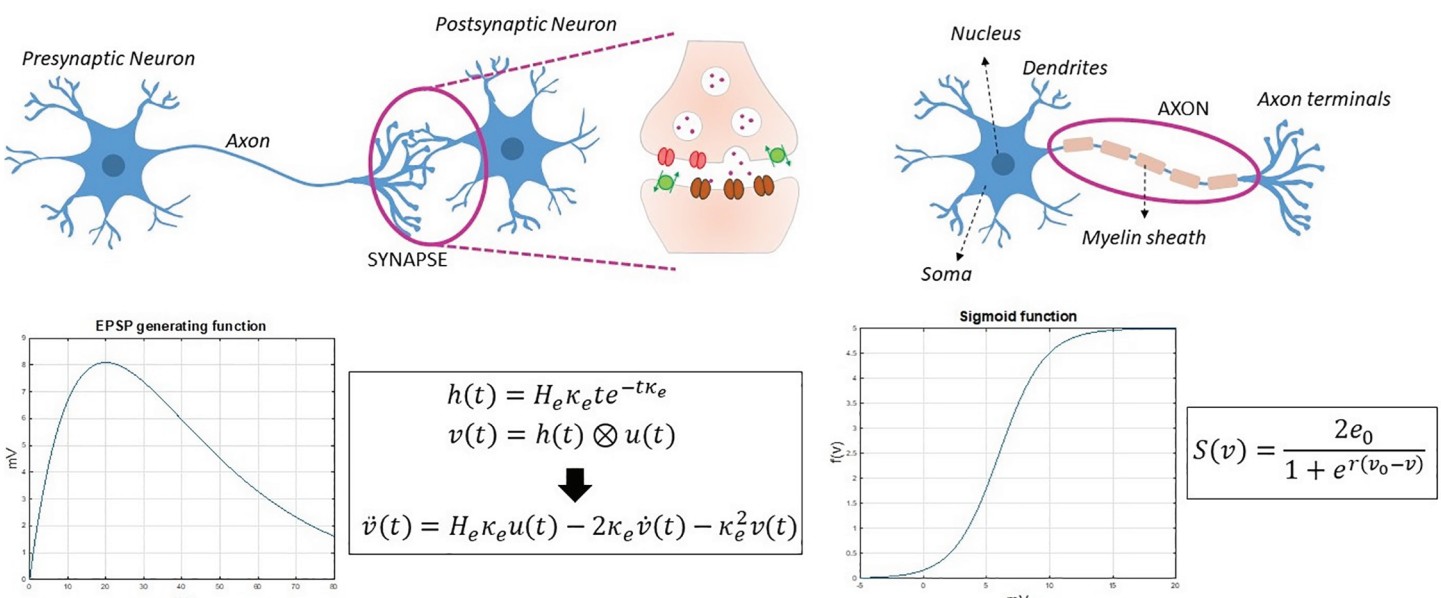

**A) Rate-to-potential operator:** dynamics between synapses and dendtritic trees

**B) Potential-to-rate operator:** produces output spike rate

$$h(t) = H_e \kappa_e t e^{-t\kappa_e}$$
$$v(t) = h(t) \otimes u(t)$$
$$\ddot{v}(t) = H_e \kappa_e u(t) - 2\kappa_e \dot{v}(t) - \kappa_e^2 v(t)$$

$$S(v) = \frac{2e_0}{1 + e^{r(v_0 - v)}}$$

**Fig 3. Foundational components of convolution-based NMMs.** Neural populations are composed of (**A**) A rate-to-potential operator describing the postsynaptic potential generated by the firing rates of the presynaptic neurons; and (**B**) a potential-to-rate operator, typically expressed as a nonlinear function, to relate the membrane potential of the neurons to their spiking activity.

stable points, yielding analytic versions of the model equations amenable to stability analysis and large parameter space exploration, has been extremely useful in understanding the dynamics of these systems and their implications for brain organization [13,15,17,72].

Even though convolution-based NMM share a common framework with two core operators, three key factors sets the models apart: (1) the number of neural population modelled, (2) the degree of physiological complexity associated with each neural population, and (3) the connectivity between them. By exploring these differences, we aim to uncover how each model uniquely captures the dynamics of alpha oscillations, providing insights into the description of the underlying mechanisms.

## Materials and methods

### Alpha rhythm models

With the basic conceptual and mathematical background established, the four selected NPMs representing alternative theories for the genesis of alpha activity—JR, MDF, LW, and RRW—will now be introduced in full detail. In the next few sections we present for each model (i) topological and circuit diagrams with the corresponding equations, (ii) alpha rhythm simulations with numerical expressions, and (iii) a didactic commentary. By comparing and contrasting these models in the subsequent sections, we aim to provide insights into their activity regimes and dynamical properties. All model parameter definitions, values, and units are listed in S9 Appendix. Selected equations are included in figures to assist exposition, while the complete equations for all models can also be found in S9 Appendix, as well as in the Python code implementations in the GitHub repository accompanying this paper:

(https://github.com/GriffithsLab/Bastiaens2025_AlphaModels). Regarding nomenclature: originally we aimed to find a generalized mathematical form that covered all four models of interest (as in e.g. [66]), and allowed for a single and consistent set of symbols with clear correspondences across models indicated by variable and parameter names. After further exploration we determined however that this is not possible without an unhelpfully large amount of abstraction. We have therefore elected to write out the equations exactly as they appear in the original and/or primary literature sources, which are also in most cases the conventional terms used in the literature and current practice.

**Jansen-Rit model.**   Based on Lopes da Silva's lumped parameter formulation [72], the JR model was one of the first of its kind to reproduce a broad range of EEG oscillation frequencies (including alpha), as well as evoked response waveform, by describing the macroscopic electrophysiological activity within a cortical column [9,73]. Analogously to Zetterberg et al. [74], JR developed the model with three interconnected neural populations: pyramidal projection neurons ($y_0$), excitatory ($y_1$) and inhibitory ($y_2$) interneurons forming two feedback loops—a (fast) excitatory feedback loop and a slow inhibitory feedback loop (Fig 4A; [75]). The output $y_1 - y_2$ represents the net PSP on the pyramidal cell dendrites, which is defined as the difference between the EPSP from the excitatory population and the IPSP from the inhibitory population. This quantity corresponds to the membrane potential of pyramidal neurons, which can also be understood as the output of the columnar microcircuit that is transmitted to other adjacent and distal brain areas. Since pyramidal neurons have their apical dendrites in the superficial layers of the cortex where the postsynaptic potentials are summated, their activity is the primary contribution to the measured EEG signal [9,76].

The mathematical expression of the sigmoid for JR is defined as

$$S(v) = \frac{2e_0}{1 + e^{r(V_0 - v)}} \tag{1}$$

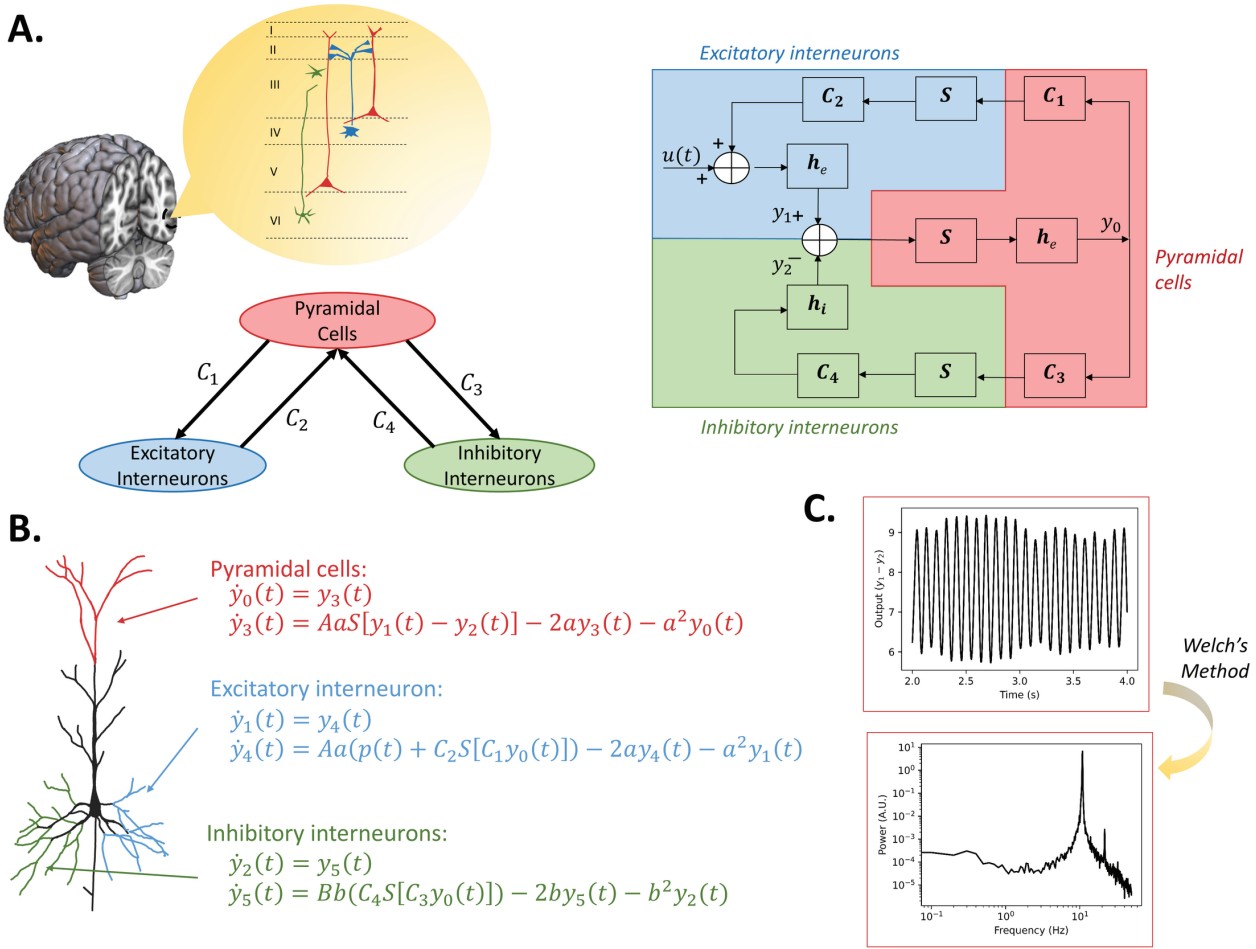

**Fig 4. JR model topography, schematic, numerical mathematical expression, and alpha simulation results.** (**A**) General structure of the model, along with a detailed schematic that includes the operators and representations of the connectivities. (**B**) Numerical mathematical expression for each neural population (neuron hand-drawn based on Fig 1 in [45]). (**C**) Simulation outputs of the model with standard parameters (time series, power spectrum estimated from the time series).

with $e_0$ representing the firing rate at threshold (and $2e_0$ the maximum firing rate), $r$ denoting the variance of firing thresholds, and $V_0$ the mean firing threshold. The impulse response is expressed as follows

$$h(t) = \alpha\beta t e^{-\beta t} \qquad \text{for t > 0,} \qquad (2)$$

The parameter $\alpha$ is defined as the maximum amplitude of the postsynaptic potential, and $\beta$ represents a sum of the reciprocal of the time constant of the passive membrane and all other spatially distributed delays present in the dendritic network, condensed into a single lumped term. $\alpha$, $\beta$ in Eq. 2 correspond to the terms $A$, $a$ and $B$, $b$ in Fig 4 for the excitatory and inhibitory populations, respectively.

After transforming the impulse response in the Laplace domain, the system is fully defined by second-order differential equations (see S1 Appendix). These equations are detailed in Fig 4B, along with the numerically integrated time series output, and the associated power spectrum in Fig 4C. The connectivity parameters $C_1$ and $C_3$ differ slightly from $C_2$ and $C_4$. JR assumes pyramidal cell populations synapse equally onto two other populations, but the

synaptic coefficients at the dendrites of the excitatory and inhibitory populations differ. Conversely, the synaptic coefficient at the dendrites of pyramidal cells is fixed, while their synaptic connectivity changes. Thus, technically speaking, $C_1$ and $C_3$ represent synaptic coefficients, and $C_2$ and $C_4$ are connectivity constants [44]. Nevertheless, in practice all four $C$ parameters do represent connectivity strength within the columnar circuit, and can and are evaluated as such. Further details on JR equations are give in S9 Appendix.

**Moran-David-Friston model.** Many models inspired by JR emerged in the years following their introduction. One of the most influential of these was proposed by David and Friston [12], later extended by Moran et al. [15]. The MDF model and the JR model (of which it is an indirect extension) thus share many similar features, and are interesting to compare in terms of the new elements included in [12] and [15]. One such element is the addition of recurrent inhibitory connections, which were introduced by Moran et al. [15] in order to enable the generation of a wider range of oscillatory frequencies. Another is that the contribution from excitatory and inhibitory populations are separated in the equations, giving rise to independent EPSP and IPSP terms. The quantity used in observation models such as EEG as a measured response corresponds to the difference between these two postsynaptic potentials, resulting in supplementary sets of differential equations. A third main modifications from JR in MDF is the expression of the sigmoid, given by

$$S(v) = \frac{1}{1 + e^{-\rho_1(v-\rho_2)}} - \frac{1}{1 + e^{\rho_1\rho_2}}. \tag{3}$$

This differs from the other models surveyed in this paper (cf. Eqs 1, 4, 7) in providing a greater flexibility in its gain behaviour, parameterized by shape and position $\rho_1$ and $\rho_2$.

The impulse response in MDF is identical to JR, and the parameters have the same definition (Table B in S9 Appendix) with some small variable name changes ($\alpha,\beta = H_e, \kappa_e$ for the excitatory populations, and $\alpha,\beta = H_i, \kappa_i$ for the inhibitory population). The schematic, detailed equations and output of the model are shown in Fig 5.

**Liley-Wright model.** Liley et al. [10] developed a physiologically parametrizable, two population firing-rate based model of EEG/ECoG dynamics, which differs from JR and MDF in several respects. Most notably, this includes (i) inclusion of high-order excitatory and inhibitory neurotransmitter kinetics, (ii) presence of synaptic reversal potentials, and (iii) the separation of each neural population into both a dendritic and a somatic compartment, yielding two membrane potential state variables per population instead of one. The LW model can be thought of as a convolution-based NPM with conductance-based synaptic dynamics (where a neuron is regarded as an electrical circuit and the membrane response follows the inflow and outflow of current through ionic channels). These additional features make it more physiologically realistic than, e.g., JR, MDF, and WC, albeit at the expense of greater levels of complexity and nonlinearity [44]. As with the RRW model discussed below, LW was initially formulated as a macroscopic neural field model, with both spatial and temporal variation in the excitatory and inhibitory neural population equations. The version presented here is simplified, however, by neglecting spatial components (setting partial derivatives in the spatial terms of the original equations), and only considering the temporal dynamics—which nevertheless preserves the essential qualitative behaviour (alpha-frequency fluctuations) that is our focus in the present paper. These expressions are based on the presentations by Song et al. [77] and Hartoyo et al. [13], in which LW was used to explore periodic discharges in acute hepatic encephalopathy and eyes-open/closed alpha-blocking, respectively.

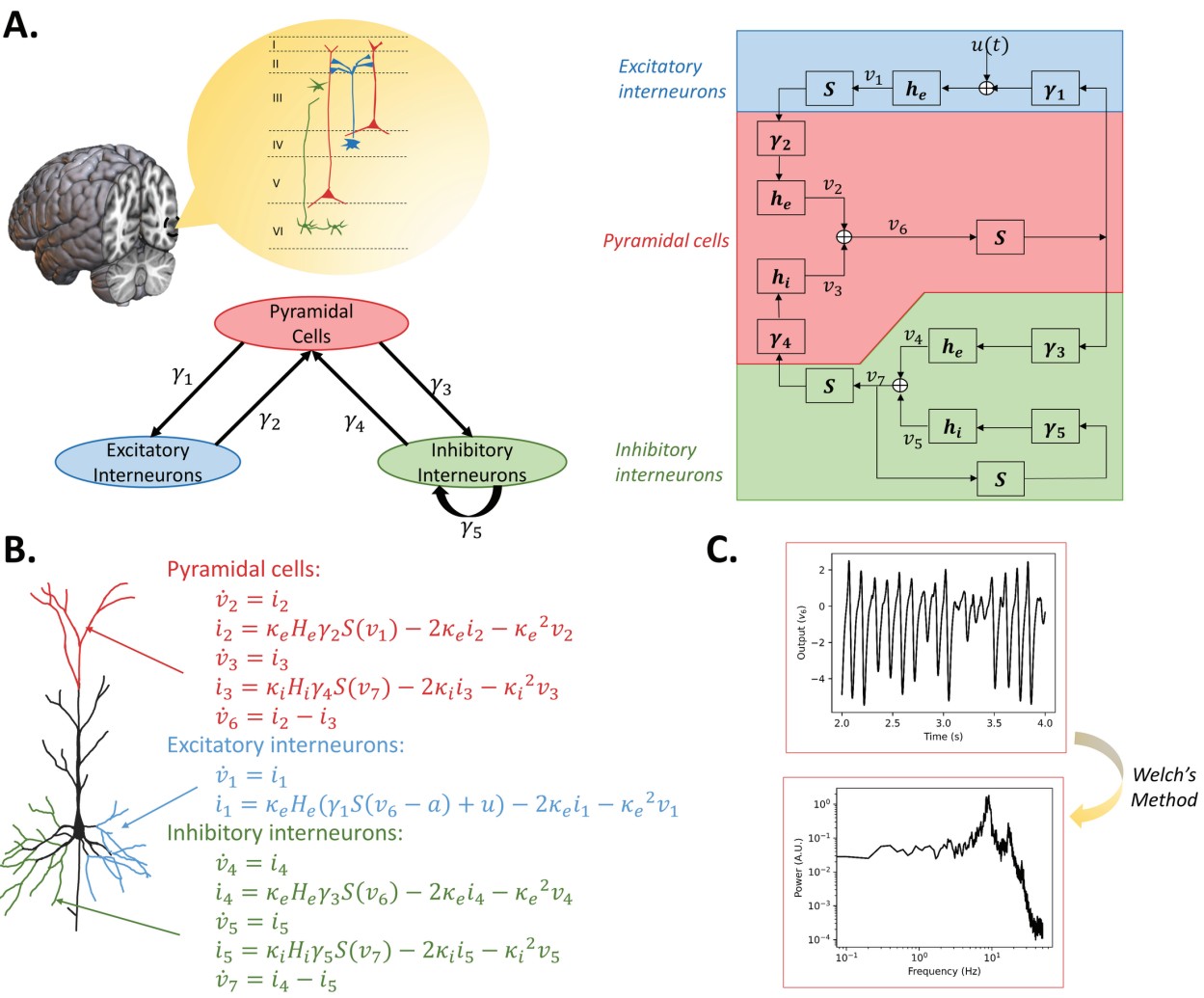

**Fig 5. MDF model topography, schematic, numerical mathematical expression and alpha simulation results.** (**A**) Composed of three neural populations with similar wiring structure to JR with the addition of an inhibitory self-connection. (**B**) Numerical mathematical expression for each neural population (neuron hand-drawn based on Fig 1 in [45]). (**C**) Simulation outputs of the model with modified parameters to generate alpha oscillations (time series, power spectrum estimated from the time series).

The sigmoidal firing rate function in LW is defined as

$$S(t) = \frac{S^{max}_{(e,i)}}{1 + e^{-\sqrt{2}(V(t)-\mu_{e,i})/\sigma_{e,i}}} \tag{4}$$

where $S^{max}_{(e,i)}$ corresponds to the maximal attainable firing rate, $\mu_{e,i}$ is the spike threshold, and $\sigma_{e,i}$ is the corresponding standard deviation. The soma membrane potential is given by

$$\tau \dot{V}(t) = V^r - V(t) + \sum \psi(V(t))I(t) \tag{5}$$

where

$$\psi(V(t)) = \frac{[V^{eq} - V(t)]}{|V^{eq} - V^r|},$$

with $V_r$ as the mean resting membrane potential, and $V_{eq}$ the mean equilibrium potential. Similarly to MDF and JR, the impulse response in LW is expressed with an alpha function,

$$h(t) = \Gamma \gamma t e^{1-\gamma t} \qquad \text{for } t > 0 \qquad (6)$$

with a postsynaptic potential peak amplitude $\Gamma_{e,i}$ and rate constant $\gamma_{e,i}$. Schematic, detailed equations and corresponding output of the model are presented in Fig 6.

**Robinson-Rennie-Wright model.** Unlike the three models discussed thus far, the RRW model does not attempt to offer a minimal circuit representation of a single cortical macro-column. Instead, this model includes thalamic neural populations in addition to cortical ones, and thus is primarily concerned with describing cortico-thalamic interactions. RRW permits the exploration of the second class of alpha theory outlined in Fig 2B, which hypothesize that the corticothalamic loop is central for resting state alpha. The model consists of four neural populations [17], two cortical (excitatory and inhibitory, similar to previous schematics) and

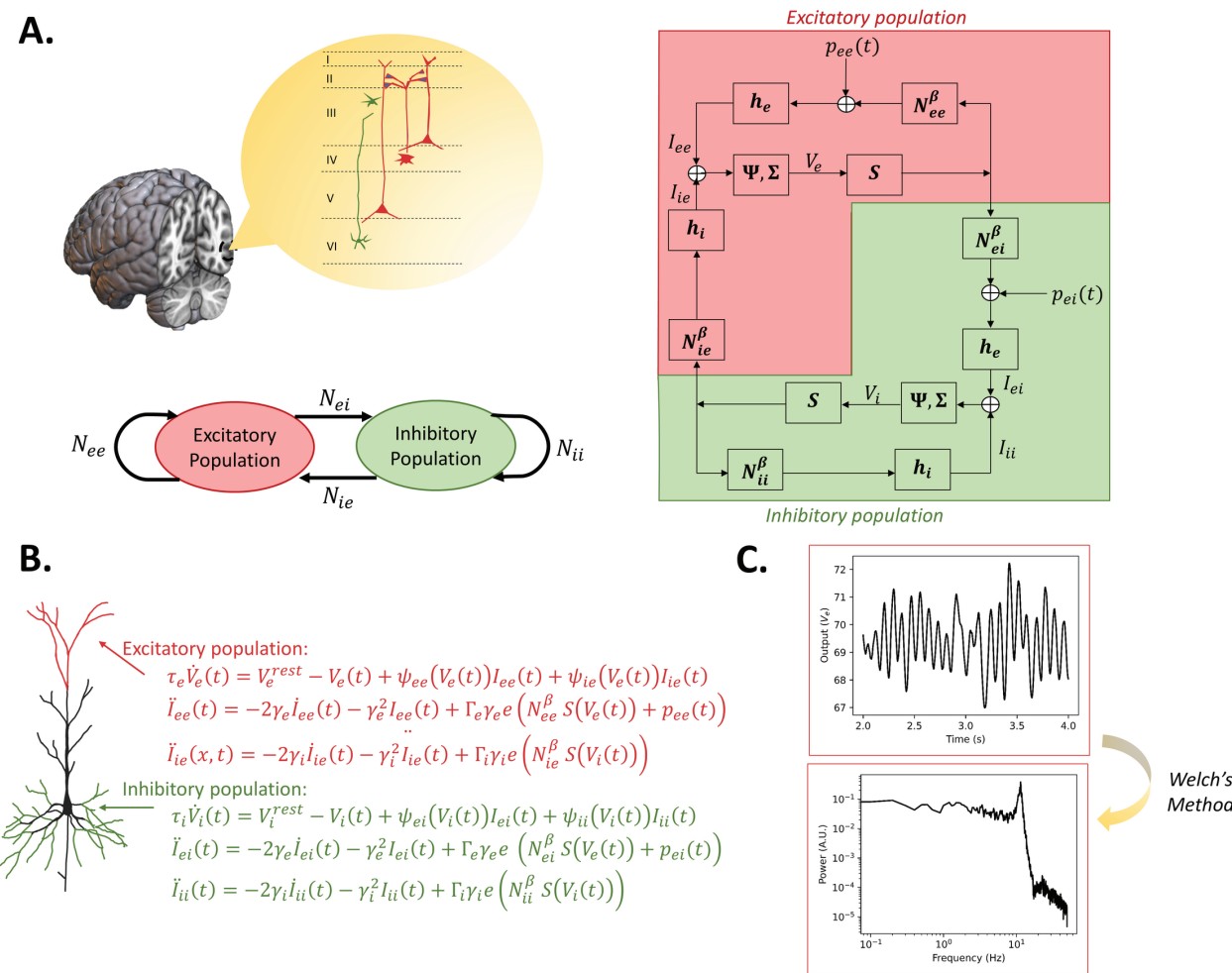

**Fig 6. LW model topography, schematic, numerical mathematical expression, and alpha simulation results. (A)** The general structure of the model is two neural populations each with a self-connection. In the detailed schematic, compared to the other models, a third block is introduced to transform PSP into soma membrane potential. **(B)** Numerical mathematical expression for each neural population (neuron hand-drawn based on Fig 1 in [45]). **(C)** Simulation outputs of the model with standard parameters (time series, power spectrum estimated from the time series).

two thalamic (thalamic reticular nucleus and thalamic relay nuclei). In this case, the two cortical populations are lumped together by assuming that intracortical connections are random, making their number proportional to the number of available synapses, and implying that cortical excitatory and inhibitory voltages are equal [78]. As noted above, like LW the original formulation of RRW is as a neural field model, making use of a damped wave equation operator for including a spatial representation. However, here we again assume spatial uniformity, removing any spatial variations, as indeed is commonly done in analyses of this model (e.g. [14,17,79–81]). Propagation effects and long axonal ranges are still preserved solely for the cortical excitatory population, this being the only population large enough with distant connections for wave propagation to have a significant effect [82]. Furthermore, a corticothalamic loop delay parameter ($t_0$) is introduced to take into account the conduction delay of the signal as it passes along corticothalamic and thalamocortical axonal projections. The differential equations comprising RRW version we use here are given in Zhao and Robinson [83], who also introduced modifications to study epileptic seizures and bursting dynamics, which we omit here for clarity. The firing rate is defined as

$$Q_a = \frac{Q_a^{max}}{1 + e^{-\frac{V_a - \theta_a}{\sigma_a'}}} \qquad (7)$$

with $Q_{max}$ representing the maximum firing rate, $\theta_a$ the mean firing threshold, and $\sigma_a' \pi \sqrt{3}$ the standard deviation of the threshold distribution. The damped wave equation governing long-range axonal activity propagation is expressed as

$$D_a \phi_a = Q_a \qquad (8)$$

with $\phi_a$ corresponding to the mean density of outgoing spikes produced by population $a$, and the spatio-temporal differential operator

$$D_a = \frac{1}{\gamma_a^2} \frac{\partial^2}{\partial t^2} + \frac{2}{\gamma_a} \frac{\partial}{\partial t} + 1 - r_a^2 \nabla^2$$

In the spatially uniform case where $\nabla^2 = 0$, owing to the short range of cortical inhibitory axons and the relative smallness of the thalamus, $\gamma_a$ is so large that the approximation $\phi_a = Q_a$ can be made for $a = i, r, s$. This is called the *local interaction approximation*, and is not assumed for $\phi_e$ as the propagation effects are significant only when considering the axons of the excitatory cortical neurons, which have significantly longer length distributions [17,84,85].

The impulse response in RRW includes both synaptic rise time $\beta^{-1}$ and synaptic decay time $\alpha^{-1}$ parameters, and is defined as

$$w(u) = \frac{\alpha \beta}{\beta - \alpha} (e^{-\alpha u} - e^{-\beta u}) \quad \text{for } \beta \neq \alpha$$
$$w(u) = \alpha^2 u e^{-\alpha u} \quad \text{for } \alpha = \beta \qquad (9)$$

which is identical to the JR impulse response function (Equation 2) when $\alpha = \beta$, and implies that the differential equation form for the dendritic response is

$$D_{\alpha\beta} = \frac{1}{\alpha\beta} \frac{d^2}{dt^2} + \left(\frac{1}{\alpha} + \frac{1}{\beta}\right) \frac{d}{dt} + 1 \qquad (10)$$

In the spatially uniform case, the impulse response appears as

$$D_{\alpha\beta}V_e(t) = v_{ee}\phi_e(t) + v_{ei}\phi_i(t) + v_{es}\phi_s(t - t_0/2) \tag{11}$$

$$D_{\alpha\beta}V_r(t) = v_{re}\phi_e(t - t_0/2) + v_{rs}\phi_s(t) \tag{12}$$

$$D_{\alpha\beta}V_s(t) = v_{se}\phi_e(t - t_0/2) + v_{sr}\phi_r(t) + v_{sn}\phi_n(t) \tag{13}$$

The details of the equation for each neural population and the output of the model are represented in Fig 7.

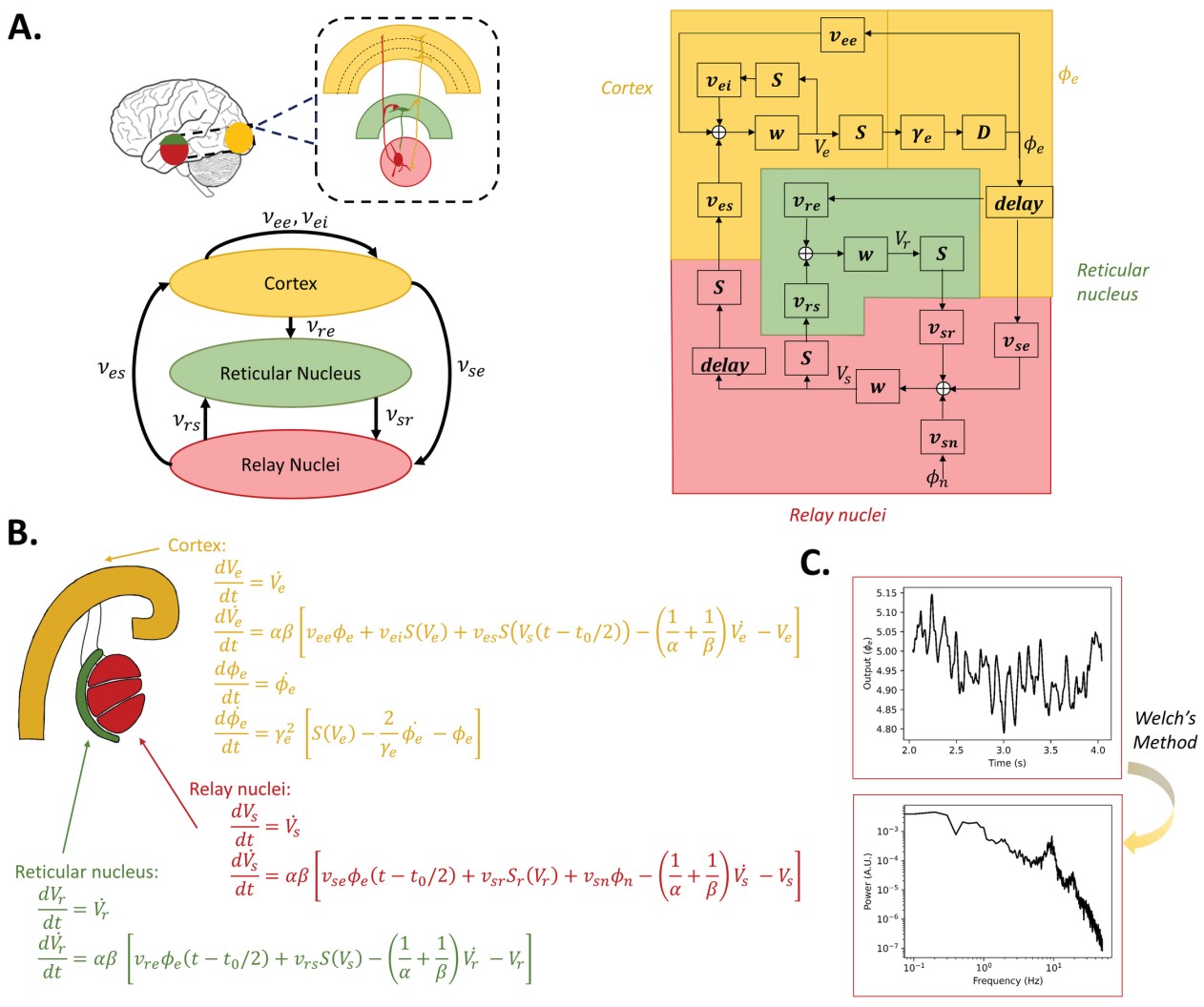

**Fig 7. RRW model topography, schematic, numerical and analytical mathematical expression, and alpha simulation results.** (**A**) Three main populations are broadly described: the cortex (composed of excitatory and inhibitory neurons) and two thalamic populations (reticular nucleus and relay nuclei). Delays are included to take into account long range connections from the cortex to the thalamus. (**B**) Numerical mathematical expression for each neural population; Numerical mathematical expression for each neural population. (**C**) Simulation outputs of the model with standard parameters (time series, power spectrum estimated from the time series).

## Simulation, power spectrum, and stability analysis methods

For all four of the selected models, we simulated alpha activity numerically by integrating the models' differential equations given in Figs 4–7 and S9 Appendix, and analytically, by algebraically calculating the power spectrum from the models' transfer function. Numerical simulations were run for a duration of 100 seconds, generating a time series that represents neural activity within the principal excitatory cortical population. The power spectrum of this simulated activity was then computed using Welch's method, as implemented in the scipy library [86]. We selected parameter values commonly used in previous studies to study alpha activity, which we refer to as 'standard alpha parameters': [9] for JR, [15] for MDF (using [12] to tune to a dominant frequency of alpha [8–12 Hz] instead of beta [12–20 Hz]), [10] for LW, and [83] for RRW (who in turn followed from [17,87]).

For present purposes we focused on three reference features of empirically-measured EEG activity: (i) alpha peak frequency, (ii) $1/f^\beta$ ($\beta \approx 1 - 2$) power spectral scaling [88], and (iii) the phenomenon of *alpha blocking*—attenuation of the alpha frequency peak during the transition from eyes-closed (EC) to eyes-open (EO) state. Each model's replication of these features was compared against reference values, taken in this case from empirical data [88]. The exponent $\beta$ was computed with two different methods: (1) Evaluating pre- and post-peak $\beta$ separately by fitting a line with linear regression in the logarithmic scale, and (2) Using the power spectrum fit of the FOOOF library (https://fooof-tools.github.io/fooof/); [89]), which parametrizes neural power spectra into a mixture of the $1/f^\beta$ background and a Gaussian for each frequency peak. These FOOOF fits are also used to calculate the dominant oscillation frequencies of the power spectra, which are discussed in detail in parameter space figures found in results section. To gain further insights into the dynamics generated by JR and LW, we determined the stability of the fixed points of the system as a function of E-I connection strengths, the derivations of which are given in S3 Appendix). Python (3.8) code for all signal processing and modelling analyses is available at https://github.com/GriffithsLab/Bastiaens2025_AlphaModels.

## Results

Having presented and contrasted the four candidate alpha models (JR, MDF, LW, RRW) in terms of their motivation and formulation, we now turn to an assessment of their simulated activity dynamics. First, we present numerical and analytic spectra, discussing general characteristics and comparing them quantitatively against empirical EEG features. Second, an exploration of the boundaries of the alpha regime is conducted through parameter searches, with a specific focus on discerning the impact of rate constant and connectivity on the dominant oscillation frequency. Last, a comprehensive comparison of the models is provided, encompassing various facets including their topology, mathematical equations, and the biological significance attributed to the parameters.

### Analysis of neural model dynamics

#### Characteristics of model-generated alpha activity

*Frequency peak and harmonics.*

Each of the models displays a dominant oscillatory frequency within the alpha range for the originally-reported default parameters, with values of 10.8 Hz, 8.8 Hz, 11.6 Hz, and 9.5 Hz observed for JR, MDF, LW, and RRW, respectively (Fig 8A). With these parameter settings, JR closely approximates the 10 Hz frequency, while LW demonstrates a slightly higher value, and RRW a lower value. Importantly, all of these frequencies fall well within the alpha oscillatory

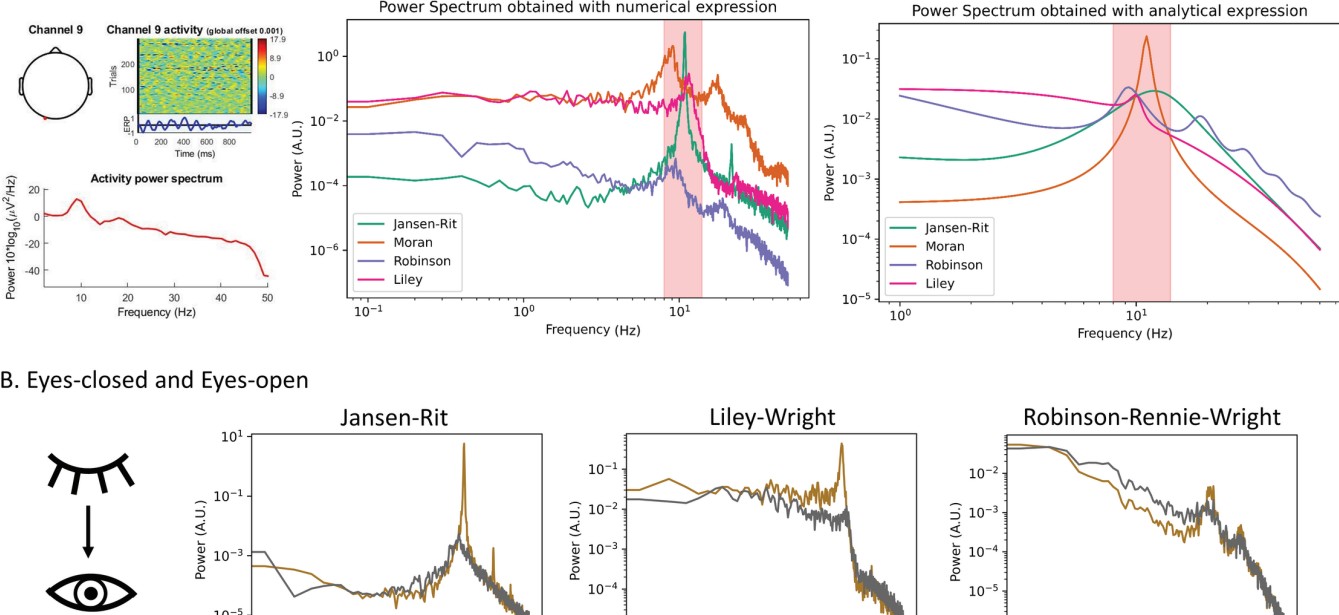

**Fig 8. Simulation results with standard parameter settings to generate characteristic resting state alpha oscillations features.** (**A**) Power spectra with characteristic occipital alpha rhythm from empirical EEG time series (left), from numerical simulation results (middle), and from analytical simulations (right). The red zone in the simulated results corresponds to the alpha range. All models generate an alpha oscillation with variations in specific features (peak frequency, presence of harmonics, 1/f shape). (**B**) Simulation results for EC and EO in JR, LW and RRW. The difference from EC to EO is an attenuation in the amplitude of the alpha rhythm.

range of 8–12 Hz, indicating that the models adequately simulate the alpha frequency peak. It should also be noted that there is considerable heterogeneity across subjects in terms of both the central frequency and magnitude of the alpha rhythm [90], and slight modifications in the model parameters have the potential to shift the peak frequency up or down, providing flexibility in matching specific experimental recordings. Differences between individuals in model parameters can be potentially also related to their cognitive profile as, alpha peak is considered as a biomarker for healthy cognitive functioning [24,91].

In addition to the main frequency, harmonics in the beta range are also present in each model, albeit with varying degrees of accentuation. Of these, LW exhibits the least pronounced harmonics, suggesting a closer approximation to a pure sinusoidal waveform. In contrast, RRW shows more prominent harmonics, which is evidenced in particular by the fact that (unlike the other three models) these still appear in its linearized approximation. For further details and discussion of the linearized model approximations and transfer function equations, see S2 Appendix. The variable presence of harmonics across the four models, and their subtle dependence on parameter values and nonlinearities, underscores the complex nature of alpha oscillations in the brain and their spectral characteristics.

*1/f scaling.*

Empirical studies have shown that aperiodic activity (also known as 1/f noise) observed in EEG power spectra following a power-law function could play a functional role in healthy

brains and explain disease symptoms. For example, cognitive decline in ageing has been associated with increased 1/f noise (slope) in the power spectrum [92], as well as aperiodic variations in stroke patients [93]. The 1/f noise is therefore an important feature of resting state EEG. Visually, the shape of the 1/f curve from RRW closely resembles the empirical 1/f curve (see e.g. [94,95]). In contrast, this feature is poorly represented by JR, which may be due to the fact that the JR system generates almost a perfect sinusoid, whereas RRW for instance seems to have more aperiodic fluctuations in the EEG time series.

Table 1 presents the computed data feature values across all four models. Comparison with the mean empirical EEG result (1.36) shows that 1/f pre-peak values are considerably lower for JR and LW (0.36 and 0.48 respectively), but higher for RRW (1.64). Empirically, lower frequencies (pre-peak) exhibit steeper slopes in frontal areas, but these quantities for the JR and LW models are notably low. At higher frequencies (1/f post-peak), JR has the steepest slope (4.03), followed by RRW (3.78) then LW (2.46). All three models yield post-peak values above the empirical mean (1.48). Inversely to lower frequencies, empirically these higher frequencies in the 1/f post-peak range tend to have steeper slopes in posterior areas. However, the simulated post-peak values observed are significantly higher than the empirical values provided in [88].

To summarize, the models demonstrate an under-representation of lower frequencies in JR and LW, and an over-representation in RRW. They all exhibit considerably steeper slopes for higher frequencies than the empirical average. This discrepancy may arise because the empirical values reflect an average across the cortex, while our models aim to capture the characteristic eyes-closed alpha peak, predominantly observed in the brain's posterior region. Visually, RRW appears to be the most similar to empirical resting state EEG, especially for the representation of 1/f in lower frequencies, which is not accounted for in the other models. Finally, consistent with empirical findings, all models have lower pre-peak 1/f values than post-peak 1/f values during EC, with higher frequencies displaying steeper slopes in posterior areas within the cortex.

*Eyes open vs. eyes closed.*

A defining characteristic of the resting state alpha rhythm in visual areas is that its amplitude is attenuated in EO compared to EC conditions, a phenomenon known as *alpha blocking* [96–98]. We examined the ability of our surveyed models to reproduce this effect by modifying relevant parameters based on previous research findings. In LW, increasing the external input to the inhibitory cortical population resulted in a reduction of alpha activity, consistent with the intuitive idea that an increase in the amount of incoming visual information is what characterizes the transition from EC to EO [99]. Similar effects were also observed in the

**Table 1. Evaluating model performance against empirical EEG features**

| Model | Main fr. | 1/f pre-peak | 1/f post-peak | Harmonics |
|---|---|---|---|---|
| JR | 10.8 | 0.39 | 4.04 | Y |
| MDF | 8.8 | 0.10 | 5.50 | Y |
| LW | 11.6 | 0.48 | 2.46 | Y |
| RRW | 9.5 | 1.64 | 3.78 | Y |
| Empirical | $\approx 10$ | 1.36 | 1.48 | Y |

To assess the performance of each neural mass model, we estimated its characteristic features, such as the main frequency, slope, and presence of harmonics, and compared them against the corresponding empirical measures obtained from resting state EEG recordings. These features are known to be informative of the underlying neural dynamics that give rise to the EEG signal. By evaluating the agreement between the model-based estimates and the empirical approximations, we can determine the extent to which the model captures the essential aspects of brain activity during rest.

JR and MDF models, where an increase in external input led to the alpha blocking. In these cases however, input is (and can only be) delivered to the excitatory rather than the inhibitory neural population. For RRW, we selected a specific parameter set that simulates the EO state based on detailed studies conducted by Rowe et al. [87]. According to these authors, the transition from the EC to EO state is associated with a decrease in cortico-thalamocortical and intrathalamic gains, accompanied by increased cortical gains and dendritic rate parameters, which together lead to an alpha blocking behaviour in RRW. Interestingly, these observations regarding RRW are broadly consistent with the behaviour of the three intracortical models: In JR, MDF, and LW, the attenuation of the alpha rhythm is caused by an increase in input representing incoming visual stimuli. In the case of RRW, it is mediated not by a direct input per se, but by a decrease in cortico-thalamic interactions and an increase in cortical gains. This increase in cortical activity causing alpha blocking in RRW could be considered analogous to the increase in cortical activity caused by greater driving input in JR, MDF, and LW.

In summary, all four models capture key features of empirically observed alpha rhythms, in terms of frequency peaks, harmonics, alpha blocking, and 1/f scaling. Of the four, RRW is in general notably closer to empirical EEG data in both its 1/f behaviour and its harmonics. It is important to acknowledge however that this analysis is based on a specific set of parameters, which can be restrictive given the wide range of parameter combinations that can give rise to the alpha regime. Therefore, further exploration of the parameter space boundaries is crucial to gain a more comprehensive understanding of the emerging behaviour and dynamics of the alpha rhythm.

**Structure of parameter space.**   Alpha oscillations are generated by non-unique parameter sets, and while there may be quantitative differences in parameter values between models, their qualitative behaviour may be similar. Next we explore alpha regime boundaries and the necessary conditions for producing a dominant frequency in the alpha range, as a function of rate constant and connectivity parameters. We also identify any other dynamical regimes that the models may present. Parameters with similar biological interpretations between the models are compared in order to provide a meaningful comparison. To ensure consistency, all other parameters are maintained in their standard resting state setting (provided in Tables A, B, C and D in S9 Appendix for JR, MDF, LW and RWW, respectively).

*Rate constant parameter space dynamics.*

The JR, MDF and LW models have distinct excitatory and inhibitory impulse responses that are modulated by rate constants ($\tau_e$ and $\tau_i$). These rate constants reflect collective passive dendritic cable delays and neurotransmitter kinetics associated with fast synaptic activity involving glutamatergic AMPA and GABA receptors [70]. This synaptic filtering is assumed to take a different shape in excitatory than in inhibitory neural populations in most of the four models, with the exception of RRW—where the same rate constant is used for AMPA as for GABA receptors. Previous studies have demonstrated that the manipulation of these rate constants can significantly impact the dominant frequency of oscillations [12,100]. In our investigation, we aim to determine whether similar patterns of frequency changes can be observed across the parameter space for all three models.

Across all models, a consistent trend is observed where the predominant rhythmic frequency decreases with an increase in both rate constants, aligning with previous analyses [12]. For LW, the range of values for $\tau_e$ and $\tau_i$ differs due to the system's tendency to diverge if $\tau_e$ becomes excessively high compared to $\tau_i$. Due to this, in Fig 9 we constrain the possible range of values to 1-10 ms for $\tau_e$ and 10-60 ms for $\tau_i$. With a uniform external input, JR has a peak oscillatory frequency of 12.4 Hz, falling within the high alpha / low beta range. MDF can elicit higher beta oscillations with a normal noise input when the rate constants are both small. This

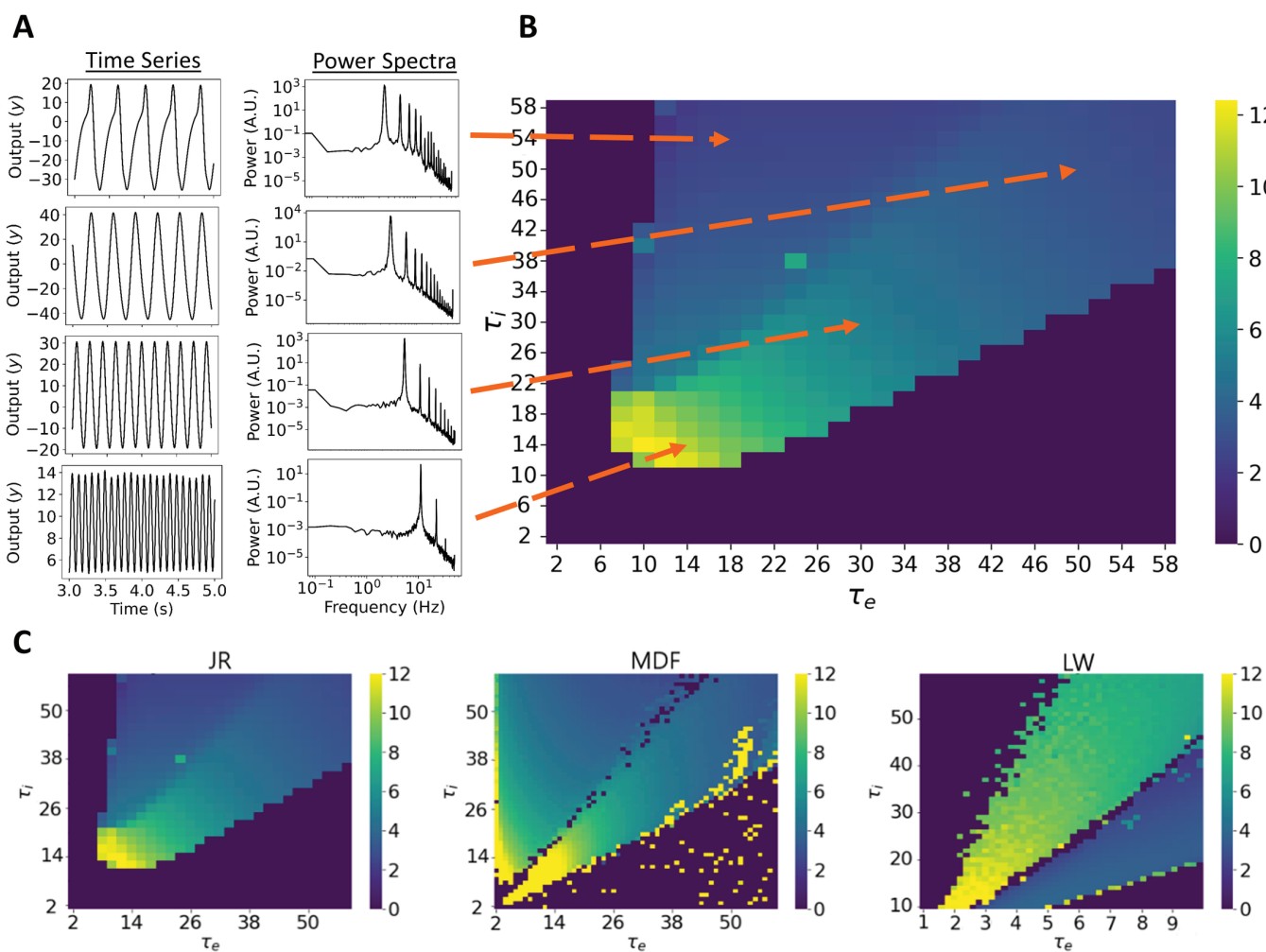

**Fig 9. Effect of rate constants on dominant frequency of oscillation for the JR, MDF, and LW models.** (**A**) Example time series and power spectra of a set of specific rate constant values to show the slowing in frequency as the values of the excitatory and inhibitory rate constant increase. (**B**) Heatmap presenting the dominant frequency of oscillation as a function of the rate constants of the JR model. (**C**) Three heatmaps for the JR, MDF and LW with the dominant frequency of oscillation as a function of the rate constants. For JR and MDF $\tau_e$ and $\tau_i$ are varied from 2 ms to 60 ms. For LW, $\tau_e$ changes from 1.72 ms to 5 ms, and $\tau_i$ from 10 to 50 ms to generate oscillatory behaviour.

suggests that the inclusion of self-inhibitory connections in MDF contributes to generating higher frequency oscillations. Notably, both JR and MDF exhibit a phenomenon known as a 'hypersignal' [12] when $\tau_i$ is considerably higher than $\tau_e$, which is typically associated with lower frequency oscillations. In such cases, the time series does not produce an exact sinusoidal oscillation (Fig 9). Conversely, if $\tau_e$ becomes too high compared to $\tau_i$, neither model shows oscillatory patterns. This means that a balance needs to be kept in order to maintain a periodic behaviour, which can be achieved by keeping the product of $H_{e,i}$ and $\tau_{e,i}$ constant by appropriately adjusting $H_e$ and $H_i$ as $\tau_e$ and $\tau_i$ is modified [12].

In LW, equivalent hypersignal behaviour is observed when $\tau_e$ is excessively high compared to $\tau_i$, while in the opposite case of $\tau_i$ higher than $\tau_e$ no oscillatory activity is seen. Furthermore, this hypersignal activity occurs above the alpha regime in $\tau_e$ vs $\tau_i$ space for JR and MDF, and below the alpha regime for LW (Fig 9). What these observations suggest is that the central alpha oscillatory regime in JR and MDF operates in a manner that is intrinsically

different to the alpha regime in LW—a question we revisit through the lens of linear stability analyses below.

As expected, modifying the shape of the synaptic filtering through the rate constants has an influence on the rhythmic behaviour of the system. Increasing both rate constants simultaneously leads to a decrease in the frequency of oscillation, as longer delays are then introduced. For example, if a disease affects the propagation of action potentials, it could lead to a decrease in the dominant frequency of oscillation. In RRW, $\tau_e$ and $\tau_i$ are assumed to be equal, considering that the difference in rise time between AMPA and GABA-A is negligible and, therefore, the synaptic filtering is the same between excitatory and inhibitory neurons. This assumption could be questioned, however, as changes in rate constants in the other models have been shown to affect the central frequency.

*Connection strength.*

The strength of connections between neural populations plays a role in facilitating communication, and thus when the strength of these connections is appropriately balanced, it enables coordinated neural activity, leading to the generation of brain rhythms. Even though on the face of it the neural populations included in the four models differ quite considerably, they all exhibit at least one common element—a principal excitatory-inhibitory ($E$–$I$) loop. The ratio of synaptic weights within that loop relates closely to the concept of 'E/I balance', a widely studied physiological phenomenon that has garnered significant attention in neuroscience in recent years [101–104]. We explored the impact of connectivity parameters on the dominant frequency of oscillation. To maintain conciseness, we exclude the connectivity parameter spaces of MDF in this section, since the patterns observed are very similar between JR and MDF, with the distinction that MDF tends to generate higher frequencies of oscillation for the same set of parameter values. A comprehensive summary of the comparison between JR and MDF can be found in S6 Appendix. Additionally, S7 Appendix includes a 4D parameter analysis for JR, encompassing all model connectivities.

JR's E-I interaction is represented by the connectivity strength between pyramidal cells and inhibitory interneurons. Since LW is only composed of one excitatory and one inhibitory neural population, the parameters of interest are the two synaptic weights connecting the two populations. Finally, for RRW, the reticular nucleus inhibits the relay nuclei and is considered the inhibitory population of the model. In this context, we consider the relay nuclei as having a central role and can be compared to the pyramidal cells in JR, as they are connected to all other populations. The excitatory-inhibitory interaction explored is then within the thalamus between the relay nuclei and the reticular nucleus. It should be noted that this interaction is not an isolated loop, because it is embedded within the larger cortex-reticular nucleus-relay nuclei loop, and so is also affected by the activity from the cortex. However, for simplicity, our focus is on the E-I interaction between the two thalamic populations.

After exploring various parameter ranges, we identified specific values that produced distinct behaviours for each model, and focused on these dynamic regimes. Results of these analyses are shown in Fig 10. As can be seen in the heatmaps, we observe an inverse diagonal relationship between E-I connectivity and the parameter regime giving rise to alpha frequency oscillations in all three models. This illustrates the fact that it is the total amount of E-I connectivity, or the total E-I gain, that defines the presence of alpha rhythm in these models. A second common feature across all three models is that if the excitatory or the inhibitory connectivity is too low, non-physiological results are obtained. These include time series with either very low amplitude or very high frequency (dark region in Fig 10D), highlighting the importance of the interaction between these two populations for the generation of rich neural dynamics.

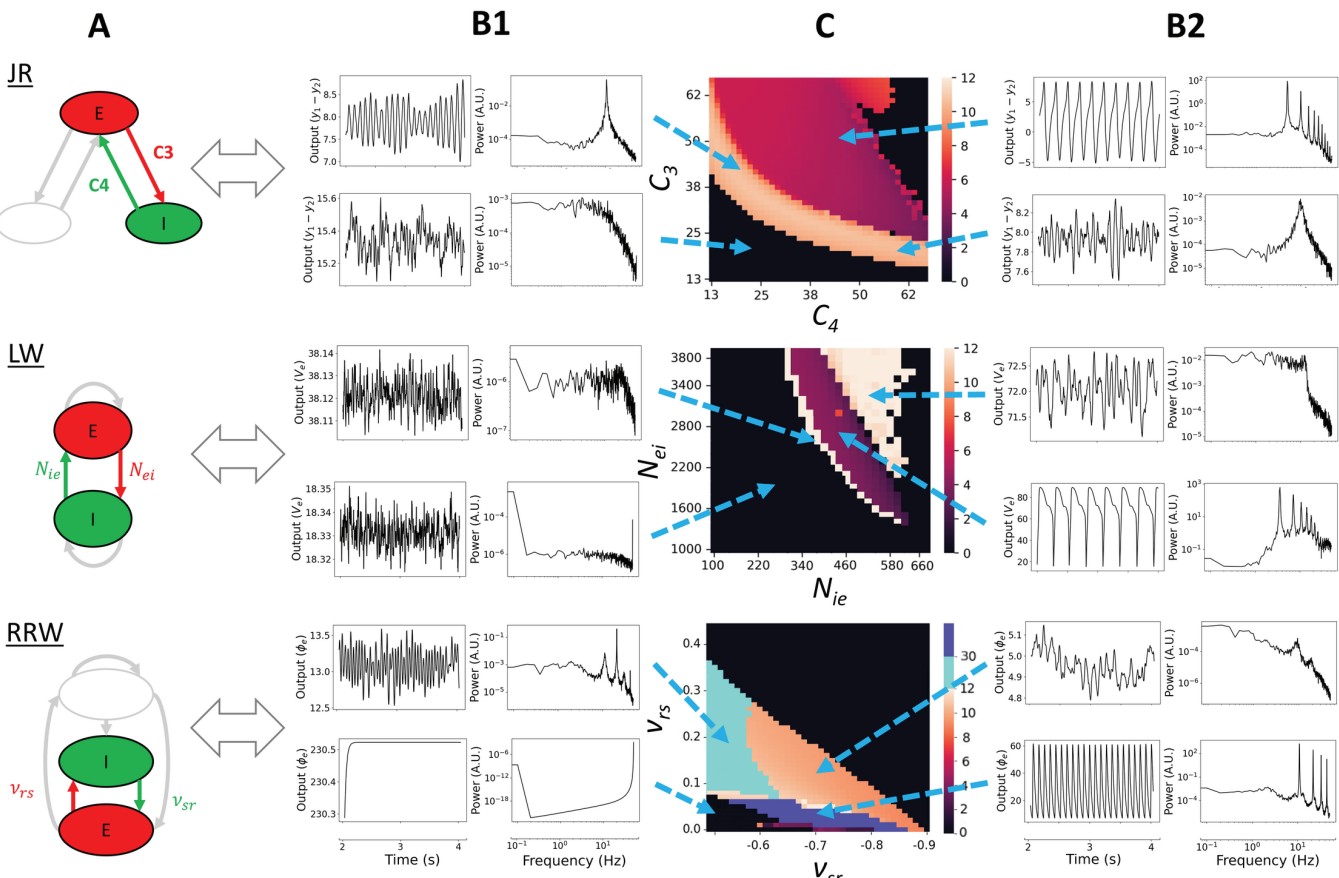

**Fig 10. Frequency of oscillation parameter spaces as a function of E-I connectivities.** (**A**) Schematic of the models with their principal E-I loop highlighted. These are the parameters that are going to be varied. (**B1** and B2) Time series and corresponding power spectra for specific combinations of E-I, showing different dynamics. (**C**) Heatmaps presenting the dominant frequency of oscillation as a function of E-I connectivity. The dark region presents non-oscillatory or non-physiological time series. JR and LW have a clearly defined regime of lower frequency of oscillations being generated (purple and red region), whereas RRW quickly tends to produce signals of lower amplitude, or higher frequency of oscillations. In RRW, the dark blue regime indicates that the system is still oscillating but at a higher amplitude and higher frequency as the system is starting to explode. In the light blue regime, the dominant frequency of oscillation is in the beta regime. In the three models, white or orange areas correspond to alpha or higher oscillations.

The relationship between $C_3$ ($P \rightarrow I$) and $C_4$ ($I \rightarrow P$) in JR needed to generate alpha oscillations follows an exponentially decaying shape. A similar correspondence is observed in LW, although with a narrower range of possibilities due to model constraints. LW also shows a steeper slope, indicating a stronger effect on oscillatory frequency of the input from GABA interneurons ($N_{ie}$) than the input to GABA interneurons ($N_{ei}$). Both the JR and LW models generate lower frequency oscillations, corresponding to the hypersignal regime, as observed in the analysis of rate constant parameter space (purple color in the JR and LW heatmaps in Fig 10C, rows 1 and 2). In the LW, if the connectivities are increased beyond this regime, predominantly alpha-frequency activity is generated (triangular white zone above the purple region), which corresponds to the dynamics observed with standard connectivity parameter values. To better understand this difference, a local stability analysis was performed to define the fixed points of the JR and LW models, and expand on their dynamical characteristics (Fig 11). In the case of JR, the coloured alpha regime presents unstable fixed points that

continue into the hypersignal regime. These oscillations are due to an Andronov-Hopf bifurcation, wherein the system enters a limit cycle that changes shape over time (Fig 11A1, 11A2). In LW, an Andronov-Hopf bifurcation also occurs, explaining the hypersignal and some higher frequencies on the left hand side of the lower frequency region (Fig 11B1 and 11B2), including alpha. However, the alpha regime in LW generated with standard parameter values lies within the space of stable fixed points (Fig 11, star in B2), which corresponds to the triangular white regime in the LW heatmap (Fig 10, C LW). This implies a separate emergent mechanism of alpha rhythm in LW that is distinct from the emergence of a limit cycle that is seen in JR. The generated alpha in this setting is noise-driven, since without noise the system becomes a damped oscillator (due to its having complex eigenvalues with negative real part), and eventually reaches the fixed point. The noise fluctuations repeatedly push the system away from its fixed point at the frequency of alpha, but it tends to stay around that stable point instead of reaching a self-sustaining limit cycle oscillation. This is shown in S4 Appendix, which includes the stability analysis of JR and LW under conditions of low or no noise input. Additionally, Fig 11 depicts the phase plane, showing only the outputs of the pyramidal cells and inhibitory interneurons. In S5 Appendix, we extend this by presenting the phase plane in 3D, incorporating the activity of excitatory interneurons as well. The stability analysis presented here corroborates the idea that the standard alpha rhythms generated by the LW and

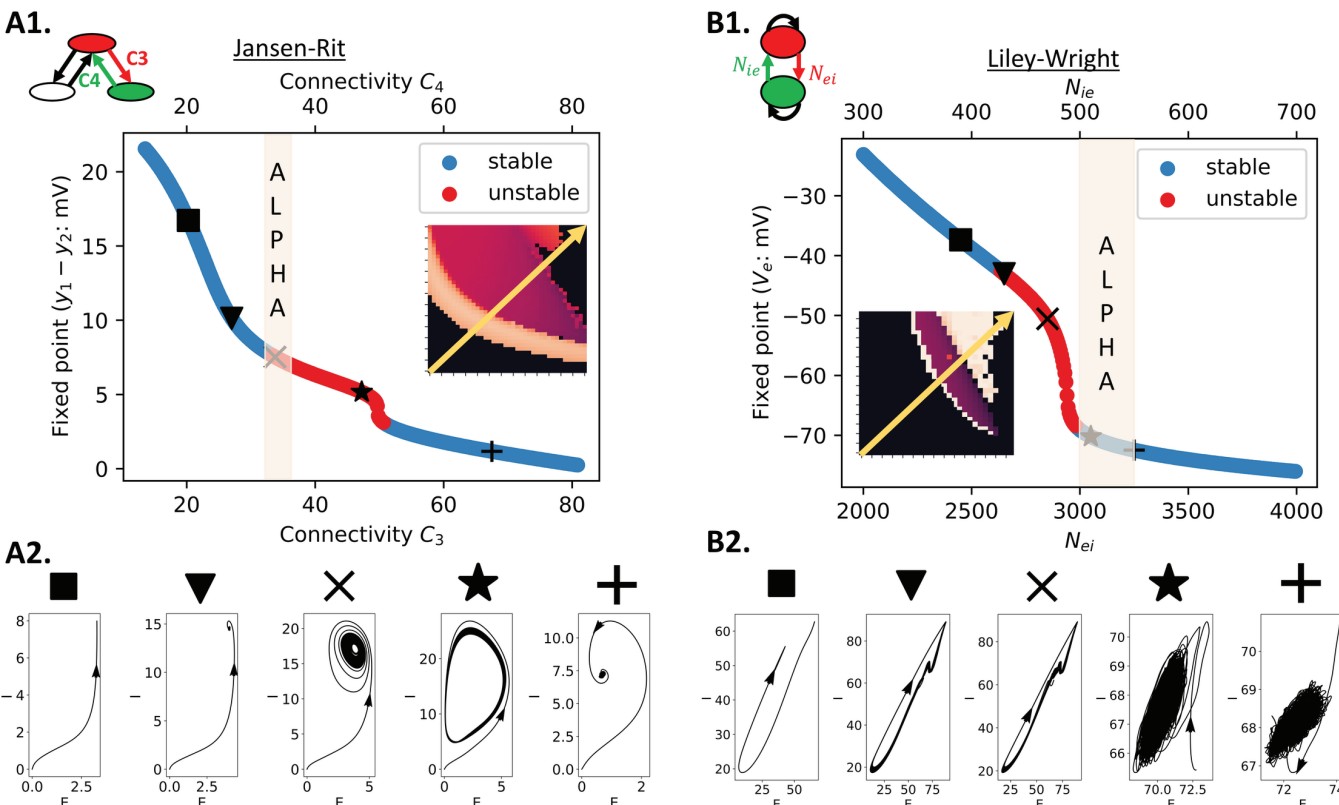

**Fig 11. Fixed points and corresponding phase planes of JR and LW at specific connectivity values with high and low noise.** By performing stability analysis, the stability of the fixed points of JR and LW is determined for connectivity values intersecting across the parameter space (yellow arrow). For JR, (**A1**) are the fixed points and (**A2**) is the phase plane for specific values of connectivity. Similarly to JR, in (**B1**) the fixed points of LW are presented with the corresponding phase plane in (B2). Unstable fixed points are red, whereas stable fixed points are blue. The light orange area corresponds to the optimal connectivity parameter setting to generate alpha oscillations in each model.

JR models constitute two mechanisms that are both physiologically and mathematically distinct. This is consistent with the rate constant and connectivity parameter space results as in the rate constant result, we could identify the hypersignal regime above the alpha regime for JR but below for LW, which is also seen in the connectivity parameter space result.

We note that, similarly to the rate constants analysis, $C_3$ ($P \rightarrow I$) and $C_4$ ($I \rightarrow P$) in JR have ranges of equal values, whereas in LW $N_{ei}$ is significantly larger than $N_{ie}$. This discrepancy can be attributed to the fact that in JR there is a higher level of excitatory interactivity, due to the additional connections between pyramidal cells and excitatory interneurons ($C_1$ ($P \rightarrow E$) and $C_2$ ($I \rightarrow P$)), which also have higher values than pyramidal-inhibitory interneurons.

As can be seen in Fig 10, the connectivity values of RRW are of a much smaller range compared to JR and LW, because they represent the connection strength (mean number of synapses times the strength of the response to a unit signal) in $mV$s rather than the number of synapses between neural populations. Extensive explorations of parameter spaces for this model have been conducted by several authors previously, often using a mathematically simpler reduced version that summarizes connection strengths across aggregated corticocortical, corticothalamic, and intrathalamic loops [78,81]. A notable feature of these analyses using the reduced RRW model is the finding that the parameters most strongly influencing the transition from an alpha-frequency regime to lower frequency dynamics are predominantly associated with the corticothalamic loop. The values of these corticothalamic loop parameters in turn determine the effect of variation in intrathalamic loop parameters on the dynamics. In our study, employing parameter sets corresponding to EC conditions, we observed that increasing the intrathalamic connectivities simultaneously led to a decrease in the amplitude of the alpha peak, accompanied by a slight shift in the central frequency. When the change in $\nu_{sr}$ and $\nu_{rs}$ are sufficiently high, then the alpha peak disappears, which corresponds to the dark coloured upper right corner of Fig 10, C row 3. Interestingly, similarly to the JR and LW models within the analogous parameter range, we observed in RRW an inverse relationship between $\nu_{sr}$ and $\nu_{rs}$. However, as $\nu_{rs}$ becomes more negative and $\nu_{rs}$ smaller, the alpha regime reduces. Frequency increases as well as the oscillatory regime as $\nu_{rs}$ becomes more positive. When $-\nu_{rs}$ is smaller than 0.6 we still have alpha oscillations, but there is a dominant peak in the beta range (around 20 Hz) seen in Fig 10B1 row 3 for RRW (light blue region). Finally, if $\nu_{rs}$ is below 0.09 approximately the system starts to explode, resulting in either higher amplitude and frequency oscillations (Fig 10B2 row 3, dark blue region) or in a continuous very high amplitude value that are not physiologically accurate (Fig 10B1 row 3, dark region). It seems that $\nu_{sr}$ has an effect on the frequency of the alpha peak which correlates with previous analysis that suggested the importance of cortico-thalamic interactions as $\nu_{sr}$ is part of the cortico-reticular-relay nuclei circuit. Adjusting $\nu_{rs}$ is key in order to have an oscillatory behaviour in the system emphasizing the E-I balance reflected in the other two models. However, due to the numerous connections within the model, the thalamus is probably not the sole connectivity parameter capable of having an effect on the frequency of alpha.

In summary, through our exploration of E-I connectivity parameter spaces in the preceding pages and in Figs 9–11, we have demonstrated that the emergence of alpha oscillations in numerical simulations with the JR, MDF, LW, and RRW models requires the neural circuit in question to reach and maintain a sufficient level of E-I gain, whilst also not exceeding a certain threshold amount. This finding emphasizes the importance of achieving a balance between excitatory and inhibitory activity and connectivity, as alterations in this balance can lead to pathological and/or non-physiological oscillatory patterns. The connectivity parameter space results we have shown indicate, in a mathematically explicit fashion, how dysregulation of synaptic connectivity may contribute to abnormal brain activity. Furthermore, in LW, we observed that the dynamics of the model are more strongly influenced

by inhibitory connectivity ($N_{ie}$) than by excitatory connectivity ($N_{ei}$). This suggests that an imbalance in the E-I ratio is more likely to be affected by the number or strength of synapses originating from GABAergic interneurons than glutamatergic ones, highlighting the significance of inhibitory interneurons and their synaptic connections in shaping the overall dynamics of LW. Our stability analyses showed that there are distinct mechanisms underlying alpha oscillations in JR and LW. In our analyses of RRW, the intrathalamic loop was seen to primarily modulate the amplitude of the alpha peak, with little influence on the dominant frequency of oscillation. Thus, in RRW, the dominant frequency of oscillation and the overall dynamics are predominantly modulated by the cortico-thalamic loop, underscoring the significance of interactions between cortex and thalamus in driving alpha rhythms according to this theory. The narrow range of parameter values leading to alpha oscillations in RRW suggests strong interdependencies among the parameters, which need to be carefully adjusted collectively to maintain oscillatory behaviour and clearly detectable spectral peaks in model simulations.

## Comparative evaluation of models

**Topology, equations, and unified parameter table.**   Initially, we compared models within the alpha regime and explored different dynamical regimes through parameter space searches. However, explicit comparisons of model components, such as topology, equations, and parameter values have not been conducted. This section addresses these aspects to evaluate the validity and suitability of models as theories of alpha rhythm generation. Table 2 highlights biologically similar and differing parameters across the models.

NPMs typically include both excitatory and inhibitory neurons. For example, LW, with a single excitatory and inhibitory population, captures excitatory-inhibitory balance and includes synaptic reversal potentials and transmitter kinetics like fast AMPA and GABA. JR adds an excitatory population, resulting in three neural populations and reflecting Katznelson's approach to explore long-range excitatory connections [73,105]. MDF introduces an inhibitory self-connection to account for high-frequency oscillations [15], while RRW, with four neural populations, includes cortical and thalamic neurons and features complex connectivities. All models use second-order differential equations combined with a nonlinear operator for synaptic processes. JR does not separately simulate EPSPs and IPSPs for pyramidal cells, unlike MDF, which includes recurrent inhibitory connections and additional differential equations. MDF also features a richer sigmoid function definition with parameters $\rho_1$ and $\rho_2$ for voltage sensitivity and position, and adaptation currents through parameter $a$. LW is more complex due to an additional block converting postsynaptic potentials into soma membrane potential and includes fast neurotransmitter kinetics. RRW describes firing behavior using a damped wave equation for cortical excitatory populations, adding an additional $\phi_e$ term for average pulse density. Understanding the role and rationale of different parameters is essential for making models biophysically meaningful. S9 Appendix includes parameter tables and their biological meanings. While all the models share common components, MDF allows easier modulation of the sigmoid function shape, and RRW introduces corticothalamic interaction parameters. LW incorporates synaptic reversal potentials, distinguishing its dynamic transformation of postsynaptic to soma membrane potentials.

**Biological basis and rationale of parameter values.**   The systems under consideration have parameters with corresponding biological interpretations; however, the nominal values assigned to these parameters vary considerably across the models. The variation in parameter values across the models can be attributed to several factors, including differences in the

**Table 2. Common parameters across models based on their biological interpretation.**

| Common Parameters | | | | |
|---|---|---|---|---|
| Model | JR | MDF | LW | RRW |
| Firing threshold (mean) | $V_0$ | – | $\mu_{e,i}$ | $\Theta$ |
| Firing threshold variability | $1/r$ | – | $\sigma_{e,i}$ | $\sigma'$ |
| Maximum firing rate | $2e_0$ | – | $S_{e,i}^{max}$ | $Q_{max}$ |
| Maximum EPSP amplitude | $A$ | $H_e$ | $\Gamma_e$ | – |
| Maximum IPSP amplitude | $B$ | $H_i$ | $\Gamma_i$ | – |
| Rate constants | $a$ and $b$ | $\kappa_e$ and $\kappa_i$ | $\gamma_{e,i}$ | – |
| Connectivity | $C_1, C_2, C_3, C_4$ | $\gamma_1, \gamma_2, \gamma_3, \gamma_4$ | $N_{ee}^{\beta}, N_{ei}^{\beta}, N_{ie}^{\beta}, N_{ii}^{\beta}$ | $\nu_{ee}, \nu_{ei}, \nu_{es}, \nu_{se}$ $\nu_{sr}, \nu_{rs}, \nu_{re}, \nu_{sn}$ |
| **Additional Parameters** | | | | |
| Sigmoid shape | | $\rho_1, \rho_2$ | | |
| Decay and rise time | | | | $\frac{1}{\alpha}, \frac{1}{\beta}$ |
| Corticothalamic loop delay | | | | $t_0$ |
| Cortical damping rate | | | | $\gamma_e$ |
| Passive membrane decay time constant | | | $\gamma_{e,i}$ | |
| Mean resting membrane potential | | | $h_{e,i}^{rest}$ | |
| Mean equilibrium potential | | | $h_{e,i}^{eq}$ | |

Certain parameters have a similar role and a biological interpretation associated with it that is comparable between the models. The additional parameters reflect the novelty and differences proposed by each models.

experimental data used to inform the models, distinct mathematical formulations, and specific assumptions. Each model is designed to capture different aspects of neural activity and may prioritize certain features or phenomena over others. In the following section, we first examine the rationale behind the expression and parameters of the firing rate function, then the impulse response, and finally the connectivity values.

*Firing rate.*

Fig 12 shows the firing rate curves of the four models. It can be seen here that there is some variability in maximum neural firing rate parameters used, as well as the point of inflection of the curves. As mentioned in the previous section, MDF implements a different expression of the sigmoid that does not include parameters equivalent to a maximum firing rate, mean firing threshold, or standard deviation of the threshold distribution in the neural population, but instead has two parameters defining shape and position. The maximum amplitude with the current setting reaches 0.9, but can be tuned by modifying the parameters $\rho_2$. Even though the other three models have parameters with a similar biological interpretation, the values are considerably different. First, the maximal firing rate is equal to $500s^{-1}$, $340s^{-1}$ and $5s^{-1}$ for LW, RRW, and JR respectively. The difference in the order of magnitude between JR and the other two models (LW and RRW) can in part be explained by the fact that the value chosen by Jansen and Rit in their original paper is taken from Freeman (1987) [106], and is actually a dimensionless normalized parameter. This quantity is expressed without units (for details on the calculation of the maximal wave amplitude $Q_m$ see [107]), whereas both RRW and LW rely on experimentally derived average values. However, in the case of RRW,

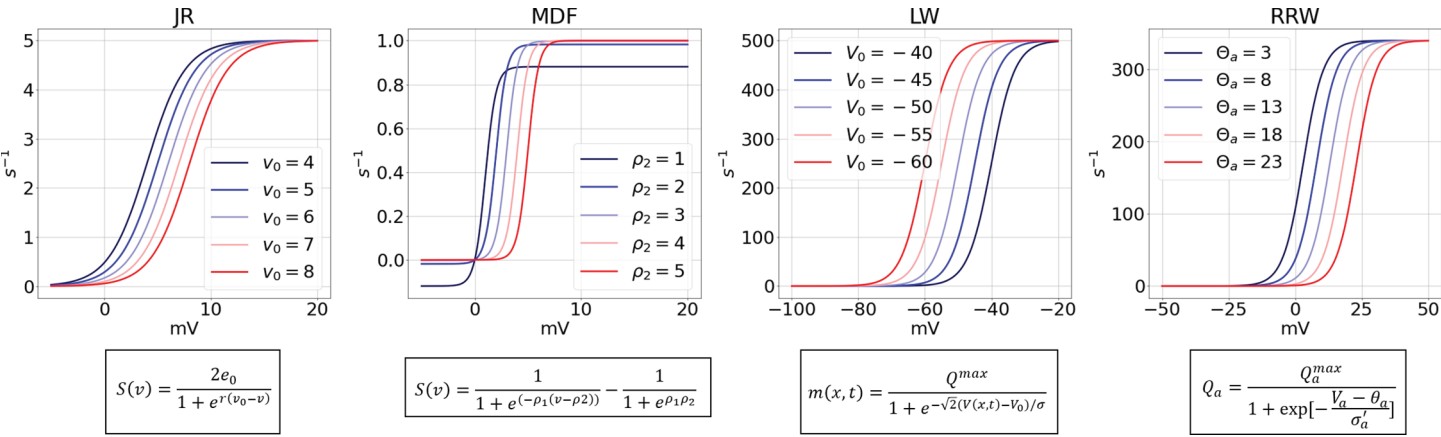

**Fig 12. Sigmoid curve of each model with firing rate against voltage with different firing threshold.** The sigmoids differ in terms of the maximum value and the voltage at which the inflection point occurs which is modulated by the firing threshold.

the assumed $Q_{max}$ value was made without a clear citation, indicating that this is an assumption, and is given in units of the maximum possible value [108,109]. The standard values from Freeman for converting membrane potential to firing rates are applied in the JR firing rate function, but the expression itself stems from Lopes da Silva et al. (1976) [110], and the current JR model uses a simplified version of that function. In the case of RRW, the firing rate function initially corresponded to the error function introduced by Wright and Liley (1995) [111]. Since 1999, the nonlinear function in RRW has been a modified version of that initial error function and closely approximates it [109]. The differing source of the firing rate conversion equation between the two models explains the slight differences observed in their mathematical expressions.

The spiking threshold parameter (voltage at point of inflection in the sigmoid curve) in LW has a negative potential, due to the fact that the model includes synaptic reversal potentials. JR and RRW, in contrast, have a positive point of inflection for this parameter ($6mV$ and $12.92mV$ respectively). The values for the standard deviation of the threshold distribution in the neural population, which affects the steepness of the firing ate slope, are $(1/0.56)mV$ ($\approx 1.79mV$), $5.5mV$, and $5.9mV$ for JR, LW, and RRW respectively.

*Impulse response.*

With respect to the impulse response, the parameter values in JR can be traced back to van Rotterdam et al. [112]. The impulse response used in JR corresponds to a simplified version of expression given in Lopes da Silva et al. [72,110]. These authors determined the parameters $A$, $B$, $a$ and $b$ by respecting certain basic properties of real postsynaptic potentials, and ensuring the system produces alpha frequency oscillations [76]. This choice of JR to use the alpha function (unrelated to alpha rhythms) as an impulse response was originally proposed by Rall [113]. MDF has an identical impulse response function, but some of the standard parameter values differ because in Moran et al. [15], the authors deliberately selected 'standard' parameters that prioritize an EEG with significant power in the higher beta frequency range, aiming to showcase the impact of nonlinearities in their computational framework. The standard MDF parameters are thus adjusted in the present study to place the central frequency in the alpha band by using comparable values to David and Friston [12]. With our adjustments to obtain alpha oscillations, the values of the impulse response in MDF vary slightly

from those in [15], such as the rate constants ($250\,\mathrm{s}^{-1}$ instead of $100\,\mathrm{s}^{-1}$ for $\kappa_e$; $62.5\,\mathrm{s}^{-1}$ instead of $50\,\mathrm{s}^{-1}$), but are still in the same order of magnitude. These differences are explained by the fact that the additional self-inhibitory connection changes the behaviour of the system for similar parameter values. Thus, to simulate an equivalent alpha these need to be modified. There is some variability across the models in the values used for EPSP and IPSP amplitudes. This has been justified physiologically by the fact that certain neuropeptides can modulate the amplitude of PSPs, meaning that some degree of freedom in choice of these values is needed [9]. For the dendritic response, the original RRW model paper [108] mentions using 'physiologically reasonable parameters' for the decay and rise rate ($\alpha$ and $\beta$), and cites sources such as Freeman [114], Lopes da Silva et al. [72], and van Rotterdam et al. [112], with no further details provided. It is surprising that the peak of the dendritic response is around 60 mV, which is considerably higher than the other models. LW, on the other hand, has a lower potential peak amplitude, which may be due the fact that other models represent the voltage at the soma, whereas LW expresses it at the site of synaptic activation [10]. One of the status intentions of LW relative to its predecessors was to be more physiologically realistic, and thus allow greater biological validity and interpretability of its parameters [10]; however it is notable that very little detail is given about the sources for chosen parameter values. Overall, an anatomical assumption made is that the amplitude of the inhibitory impulse response is larger than the excitatory impulse response, due to the fact that the former have axon terminals closer to the cell body, thereby leading to larger perturbation upon synaptic transmission [44,115]. LW makes the (reasonable) assumption that excitatory impulses occur on a faster timescale than inhibitory impulses, which is shared with JR and MDF, but notably not with RRW. In Fig 13, the shape of each model's excitatory and inhibitory impulse responses are shown, with their nominal varying rate constant values. As the rate constant increases, the curve widens and the decay time increases. In the case of RRW but not JR, MDF, or LW, variation of the decay time also leads to changes both slope and the magnitude of the impulse response curve.

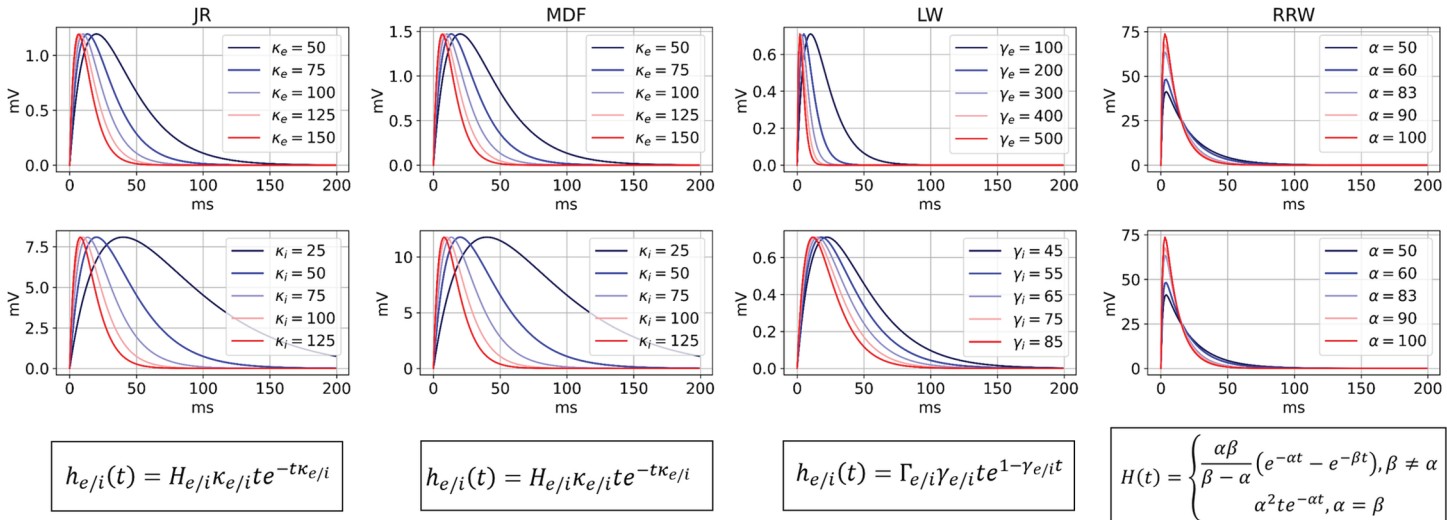

**Fig 13. Impulse response of excitatory and inhibitory population with varying rate constant.** Top: EPSP; Bottom: IPSP; except for RRW which uses the same dendritic response curve for EPSP and IPSP. The general shape of EPSP and IPSP between the models is consistent and mainly differ in terms of amplitude. Rate constant is varied for the first three models and for RRW, the different curves correspond to varying decay times.

*Connectivity.*

Connectivity parameters across the four models differ in their units and physiological interpretation, making direct comparisons of specific values challenging. In JR and MDF, the connectivity parameter values are dimensionless, and proportional to the average number of synapses between populations [9]. Based on several neuroanatomical studies [116–119] that estimated these quantities by counting synapses. With these studies, Jansen and Rit condensed the four connections into fractions of a single parameter *C* [8]. Since Jansen and Rit estimated that the global parameter *C* would most likely change primarily due to its role in capturing synaptic phenomena like neurotransmitter depletion, this reduction has been useful in determining the overall effect of variations in connectivity while keeping their proportions to each other identical. LW has parameters representing the total number of connections between the two populations, which take higher values for excitatory neurons as 80% of cortical neurons are excitatory [44]. Furthermore, anatomical estimates for each connection were derived using an equation that considers the diameter of the mean dendrite and intracortical axon, the mean total length of all dendritic and intracortical axonal arborizations, the mean length of the pyramidal cell's basal dendritic arborizations, and the neuronal density (as described in [10] and outlined in [120]). RRW has connectivity variables denoted as $\nu_{ab}$, which correspond to the mean number of synapses (anatomical or structural in nature) multiplied by the strength of the response to a unit signal expressed in units as mVs [108,109,113].

In summary, we have sought in this section to compile information from the literature on the origin of the mathematical expressions, parameter values, and biological motivations of our four models. Notable observations include: (i) even though the formulation of the firing rate curves is similar between JR, LW and RRW, their mathematical origin differs, with Lopes da Silva et al. [110] as a reference for JR, and the error function introduced by Wright and Liley [111] for LW and RRW; (ii) there is some variability across models in the parameters and equations for the impulse response; (iii) connectivity parameters can represent a proportion of the average number of synapses (JR and MDF), a total number of synapses (LW), or synaptic strengths (RRW); (iv) Although the specific parameter values may vary for the firing rate and the impulse response, modifying them uniformly yields a consistent effect across the two curves (Figs 12 and 13); and (v) similarly, as shown in Fig 10, correspondences can be made in the effects of altering connectivities.

## Discussion

### Summary of main findings

In this paper, we systematically investigated the major mathematically-expressed physiological theories of EEG alpha rhythmogenesis, focusing on four primary models (JR, MDF, LW, RRW) that cover the two main alpha theory types—intracortical and corticothalamic [40]. Our aim was to clarify the relationships between these models to prepare for future experimental and theoretical work. We examined the mathematical expression of each model, highlighting common elements and differences, and explored their parameter space to identify conditions producing alpha rhythms, focusing on rate constant and E-I connectivity strength parameters. We reported confirmatory simulation results and several novel findings. Despite differences in elements such as nominal cell types, microcircuit topologies, and connectivity assumptions, all models can reproduce the characteristic features of resting state alpha observed in empirical EEG data, with RRW better capturing the 1/f scaling and JR and LW showing more attenuated alpha blocking (Fig 8). We also examined the effect of changing the rate constant on the dominant frequency of oscillation, finding that MDF demonstrates a larger range of oscillatory behaviour. Our comparative simulations highlighted differing

positions of the hypersignal regime between JR, MDF, and LW, and demonstrated the significant impact of E-I connection strengths on model dynamics. In JR, the total connectivity strength of the inhibitory loop determines the oscillatory regime. In RRW, we observed that the intrathalamic E-I loop also plays a crucial role in modulating the general dynamics of the alpha oscillation. Decreases in inhibition lead to a dominant beta-frequency peak, and a slight shift in the alpha central frequency. However, the primary effect of the RRW intrathalamic loop (within the parameter regimes studied) was seen to be modulation of alpha peak amplitude.

Finally, we observed that changes in the number and strength of GABA interneuron synapses in LW tend to have a more prominent effect on the dynamics compared to the corresponding GABA-related parameters of the other models. Exploring the stability of the JR and LW models revealed different mechanisms for generating standard alpha oscillation: a self-sustained limit-cycle for JR and noise-driven fluctuations for LW. Our comparative evaluation highlighted topological and mathematical differences and clarified the rationales behind parameter values. Despite variations, the impact on the shape of both the sigmoid and impulse response is consistent (Figs 12 and 13). Our investigation shows similar capacities to generate spectral EEG features, leaving it unclear which alpha theory type is best supported. The selection of a model depends on the study's goal, its capacity to represent neural activity features, and relevant biological details. While mesoscopic scale empirical data may be insufficient to favour one alpha theory over another, our study clarifies the role of the E-I loop in each model and the implications of synaptic gains on dynamics, which we hope will prove useful in studies of altered dynamics associated with neural pathologies and disorders [121,122].

## Model limitations and critique

NPMs offer a valuable framework for studying brain dynamics at the mesoscopic scale using data from EEG, MEG, LFPs, ECoG, fMRI, PET, fNIRS, and wide-field calcium imaging. However, their simplicity sacrifices important neurobiological details, posing challenges for parameterization and validation, particularly in the lack of correspondence between model variables and well-defined observable quantities and neural structures [44]. Experimental validation of NPMs of this kind has to date mostly relied on human EEG data, capturing cortical excitatory neurons, leaving cortical inhibitory and thalamic populations unmeasured. Complementary data from LFPs and advanced recording technologies, such as combined electrophysiological and optical imaging in rodents, can address some of these limitations, albeit with substantial species differences. The challenges in associating model parameters with experimental observables described above also result in limitations regarding parameter translatability across models. In this work, we have therefore chosen to focus on comparing parameters of similar biological significance by analysing their relative effects—specifically, how changes from their respective baseline values influence system behaviour. While this approach provides valuable insights, the inability to make direct numerical comparisons across models, due to the significant differences in parameter scaling and baseline values, remains a major limitation. To enable such comparisons, future research could explore methods to normalize model outputs, for example, by scaling impulse response functions and sigmoid curves so that their axes and ranges align across models. Alternatively, outputs such as oscillation amplitude or frequency could be standardized against experimental benchmarks or dimensionless parameter combinations. However, developing a systematic and biologically grounded normalization framework remains an open challenge that warrants further investigation.

NPMs bridge microscopic and macroscopic brain states but involve assumptions and abstractions that may disconnect understanding across spatial scales [44,123,124]. NMMs assume uncorrelated neuron states within ensembles [47], neglecting within-population synchrony, which might impact observed EEG responses [54]. The sigmoidal function used to transform membrane potential into firing rates is a phenomenological approximation [124, 125], and individual neuron firing thresholds are thus not considered in these models.

Despite these caveats, NPMs do effectively represent brain dynamics at the meso/macro scale observed in scalp EEG, offering simplicity and computational efficiency. They enable numerical simulations, parameter estimation, and analytical correspondences for insights into physiological mechanisms [80,126,127]. In addition to limitations inherent to all NPMs, each of the four models also has its own advantages and limitations. JR has a constrained oscillatory range and requires an external drive for stable alpha oscillations, which could be considered to be inconsistent with the empirical observations that intrinsic alpha power is strongest during eye-closed states [128]. MDF includes self-inhibitory connections and spike-rate adaptation terms for higher frequency ranges, but uses parameters with limited biological relevance [15]. LW incorporates conductance-based elements like synaptic reversal potentials, enhancing neurobiological fidelity but increasing numerical instability [10]. RRW approximates EPSPs and IPSPs with the same impulse response, which has been debated, though it can reproduce empirical features like the 1/f curve and different oscillatory frequencies across brain states and neuropathologies [83,129,130].

To facilitate model comparison, our study primarily employed linear stability analysis to characterize model behaviour near fixed points and to identify bifurcation types using phase planes (such as Hopf bifurcations). This approach enabled a systematic examination of the dynamical regimes within the model. However, we acknowledge that this method does not fully account for nonlinear dynamics, such as potential chaotic behaviour, due to the approximations involved. For example, during linearization, the nonlinear sigmoid function is approximated by its slope at the operating point, effectively reducing it to a linear 'gain' parameter that reflects the population's sensitivity to small perturbations. This focus is appropriate for our study on alpha-band resting state activity, which has been consistently shown to be well-described and valid under linear approximations in previous research as operating near critical bifurcations points [87,131]. In contrast, the study of nonlinear phenomena, such as seizure states or spike-wave activity, requires different analytical tools, as these dynamics depend heavily on nonlinearity [132]. Future work could explore nonlinear behaviours more comprehensively by computing the Lyapunov exponent–a standard measure for quantifying chaos–or by simulating the model under diverse parameter settings to reveal complex temporal patterns [133–135].

A comparative analysis of these models (summarized in Table 3) reveals that the JR model shows robust global dynamics but has limitations in biological parameter significance and oscillatory range. The MDF model achieves higher frequency simulations but shares similar limitations. The LW and RRW models offer biologically associated parameters and a broad range of oscillatory frequencies, but robust global dynamics are challenging to demonstrate. The RRW model is promising for reproducing empirical features like the 1/f curve. While this table provides guidance for selecting a model based on study goals, it is important to recognize that the mathematical similarity between models makes it difficult to unambiguously identify the "correct" one for representing resting state alpha dynamics, as all can capture all or most features of empirical EEG resting state dynamics across large domains of their parameter space. Future work could address this by applying controlled perturbations—such as pharmacological interventions, transcranial stimulation, or sensory inputs—to differentiate their evoked responses. That is, for each parameter configuration of all the models studied

**Table 3. Global evaluation of the models**

| Feature | JR | MDF | LW | RRW |
|---|---|---|---|---|
| Biological significance of parameters | – | – | + | + |
| Differentiation between EPSP and IPSP | + | + | + | – |
| Oscillatory range | – | + | + | + |
| General shape of PS | – | – | – | + |
| Robust demonstration of global dynamics | + | + | – | – |
| Separation of pyramidal cells | + | + | – | – |

Different features of the models are assessed, highlighting strengths and limitations. In terms of robustness and tractability, the JR and MDF models prove more suitable. LW incorporates more physiological elements, and RRW shows a stronger capability in reproducing empirical features of alpha activity.

here, there is an associated evoked (impulse response) pattern that could be studied in conjunction with the resting (steady-state) behaviour described in this paper. Expanding analyses to include cognitive or task-based measures or combining resting state data with stimulated-state paradigms may also reveal unique, state-dependent features. These approaches offer promising pathways for resolving ambiguities and improving model evaluation.

## Conclusion and future work

In conclusion, our comparative analysis of the JR, MDF, LW, and RRW models elucidates their mathematical formulations and parameters, providing a range of biological insights. Our novel simulations showed differing precision in replicating EEG alpha characteristics, highlighting the impact of rate constants and connectivity parameters on their dynamical behaviour.

Future studies of alpha rhythmogenesis in human EEG should investigate intracortical and cortico-thalamic models at the whole-brain scale. Mesoscale data from single neural populations alone may be insufficient to distinguish between these theories. A key objective should be to determine the role of the thalamus in generating resting state alpha oscillations, and more generally to adjudicate between the cortical and cortico-thalamic alpha theories. We hypothesize that topographic variation in oscillatory brain activity, as well as network-level connectivity and dynamics, will provide important additional information for this question. Whole-brain studies must consider each node's role in the larger network, as the dynamics of neural populations may change when interconnected via the connectome. Finally, improving validation methods against empirical data, for example by extending the number and type of EEG features used for model comparison and fitting, would allow for better differentiation between models and determination of which ones offer are more accurate representation of observed brain dynamics.

## Supporting information

**S1 Appendix. Derivation of the JR model equations.** Details the derivation process leading to the formulation of the second-order differential equation used to represent the PSP output.
(PDF)

**S2 Appendix. Transfer function of the models.** Presents the transfer function equations for each model along with their corresponding outputs.
(PDF)

**S3 Appendix. Derivation of stability analysis for JR and LW.** Mathematical details of the derivation process to determine the fixed points of the system and study the stability.
(PDF)

**S4 Appendix. Stability analysis low noise JR and LW of connectivity E-I parameters.** Fixed points behaviour and phase planes of JR and LW for different parameter sets when subjected to low input noise.
(PDF)

**S5 Appendix. Phase plane of JR in 3D.** Includes the phase planes with the output voltage of the three populations under different connectivity values.
(PDF)

**S6 Appendix. Comparison of MDF and JR connectivity parameter spaces.** Analyses the effect of connectivity parameters, namely the role of the self-inhibitory loop introduced in the MDF model compared to the JR model.
(PDF)

**S7 Appendix. 4D JR connectivity analysis.** Presents the effects of the four connectivity parameters of JR simultaneously.
(PDF)

**S8 Appendix. 3D parameter space with MDF.** Reduces the 5-dimensional connectivity parameter space to a 3-dimensional representation, to assess the contributions from the circuit loops.
(PDF)

**S9 Appendix. Full model equations.** Provides the complete set of equations for the models, along with tables detailing the parameter descriptions and their standard values.
(PDF)

**S10 Appendix. Additional background literature.** Offers supplementary information on the early development of NPMs and other existing theories of alpha rhythmogenesis.
(PDF)

## Author contributions

**Conceptualization:** Sorenza P. Bastiaens, John D. Griffiths.

**Formal analysis:** Sorenza P. Bastiaens.

**Funding acquisition:** John D. Griffiths.

**Methodology:** Sorenza P. Bastiaens, John D. Griffiths.

**Supervision:** John D. Griffiths.

**Visualization:** Sorenza P. Bastiaens, Davide Momi.

**Writing – original draft:** Sorenza P. Bastiaens.

**Writing – review & editing:** Sorenza P. Bastiaens, Davide Momi, John D. Griffiths.

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
