## [Decision Letter · Decision Letter 0]

31 Dec 2024

PCOMPBIOL-D-24-01803

A comprehensive investigation of intracortical and corticothalamic models of alpha rhythms

PLOS Computational Biology

Dear Dr. Bastiaens,

Thank you for submitting your manuscript to PLOS Computational Biology. After careful consideration, we feel that it has merit but does not fully meet PLOS Computational Biology's publication criteria as it currently stands. Therefore, we invite you to submit a revised version of the manuscript that addresses the points raised during the review process.

Please submit your revised manuscript within 30 days Mar 02 2025 11:59PM. If you will need more time than this to complete your revisions, please reply to this message or contact the journal office at ploscompbiol@plos.org. Please include the following items when submitting your revised manuscript:

We look forward to receiving your revised manuscript.

Kind regards,

Jonathan David Touboul

Academic Editor

PLOS Computational Biology

Hugues Berry

Section Editor

PLOS Computational Biology

**Journal Requirements:**

At this stage, the following Authors/Authors require contributions: Sorenza Pawla Bastiaens. Please ensure that the full contributions of each author are acknowledged in the "Add/Edit/Remove Authors" section of our submission form.

3) Please ensure that the Title in your manuscript file and the Title provided in your online submission form are the same.

4) Please provide an Author Summary. This should appear in your manuscript between the Abstract (if applicable) and the Introduction, and should be 150-200 words long. The aim should be to make your findings accessible to a wide audience that includes both scientists and non-scientists. Sample summaries can be found on our website under Submission Guidelines:

5) Please upload all main figures as separate Figure files in .tif or .eps format. For more information about how to convert and format your figure files please see our guidelines:

6) We have noticed that you have cited Table  Table 2 in the manuscript file but there is no corresponding table in the manuscript.  Please amend your manuscript to include this table noting that tables should not be uploaded as individual files.

7) We have noticed that you have uploaded Supporting Information files, but you have not included a list of legends. Please add a full list of legends for your Supporting Information files after the references list.

8) Some material included in your submission may be copyrighted. According to PLOSu2019s copyright policy, authors who use figures or other material (e.g., graphics, clipart, maps) from another author or copyright holder must demonstrate or obtain permission to publish this material under the Creative Commons Attribution 4.0 International (CC BY 4.0) License used by PLOS journals. Please closely review the details of PLOSu2019s copyright requirements here: PLOS Licenses and Copyright. If you need to request permissions from a copyright holder, you may use PLOS's Copyright Content Permission form.

Potential Copyright Issues:

- Figures: 1A, 1B, 1C, 2A, 2B, 4A, 5A, 6A, and S11.. Please confirm whether you drew the images / clip-art within the figure panels by hand. If you did not draw the images, please provide (a) a link to the source of the images or icons and their license / terms of use; or (b) written permission from the copyright holder to publish the images or icons under our CC BY 4.0 license. Alternatively, you may replace the images with open source alternatives. See these open source resources you may use to replace images / clip-art:

**Reviewers' comments:**

Reviewer's Responses to Questions

**Comments to the Authors:**

Reviewer #1: Manuscript: A Comprehensive Investigation of Intracortical and Corticothalamic Models of the Alpha Rhythm

This manuscript provides a comparative analysis of neural population models (NPMs) used to simulate the alpha rhythms by focusing on four extensively studied models: Jansen-Rit (JR), Moran-David-Friston (MDF), Liley-Wright (LW), and Robinson-Rennie-Wright (RRW). The authors perform systemic comparison of the models' dynamics, including stability, parameter space, and bifurcation properties, in order to gain insights into the mechanisms underlying alpha generation.

Major Comments

1. In Section 3.2.1, the table comparing parameter values across the four models is a key strength of this study. However, the parameter values differ significantly across these models. A rational for parameter translation in the discussion section open more avenue for translatability of results across different models. As an example, a discussions subsection on how can model outputs be normalized for direct comparison could help direct future research in this field?

2. The authors heavily rely on linear stability analysis and potential nonlinear behaviors (e.g., chaotic dynamics) and their implications are not explored. While linearization is a standard tool, the omission of nonlinear dynamics could overlook some critical dynamical phenomenon. This limitation should be discussed in the manuscript. Perhaps in the discussion section.

3. The authors mention that the "Mesoscopic activity is more challenging to measure directly". However, the LFP could be used as reasonable proxy for mesoscopic neural activity as it reflects the average synaptic activity of local neuronal population. Hence, a brief discussion of how LFPs could be used as a proxy for mesoscopic neural activity could provide more context.

Typographical Errors

1. "top-down approach" is missing a hyphen in some locations.

2. Line 468 and 469: The words EC and EO are flipped in this sentence. Given that there is an attenuation of alpha activity during EO condition, the sentence should read: “….attenuated in EO compared to EC condition…”

3. Inconsistencies in references: For the reference on line 1054, the URL is written in a different font.

Conclusion

This manuscript is an impressive contribution to the modeling of alpha rhythms, offering a systematic and comparative framework for neural population models. And I believe addressing the above points will further enhance its impact and utility for the neuroscience community.

Reviewer #2: The paper is an excellent compendium of four mathematical models used for studying alpha rhythms. The authors detail the comparative advantages/features of the models in considerable detail. The study will no doubt be of value to specialists in the field. It is a pity that their analysis does not propose the experiments that would be necessary to select which of the models is the "correct" one, which weakens their work somewhat.

**Have the authors made all data and (if applicable) computational code underlying the findings in their manuscript fully available?**

Reviewer #1: Yes

Reviewer #2: Yes

PLOS authors have the option to publish the peer review history of their article (what does this mean?). If published, this will include your full peer review and any attached files.

Reviewer #1: **Yes: **Richa Phogat

Reviewer #2: No

**Figure resubmission:**
---

## [Editor Report · Decision Letter 1]

3 Mar 2025

Dear Ms Bastiaens,

We are pleased to inform you that your manuscript 'A comprehensive investigation of intracortical and corticothalamic models of the alpha rhythm' has been provisionally accepted for publication in PLOS Computational Biology.

Best regards,

Jonathan David Touboul

Academic Editor

PLOS Computational Biology

Hugues Berry

Section Editor

PLOS Computational Biology

---

## [Editor Report · Acceptance letter]

PCOMPBIOL-D-24-01803R1

A comprehensive investigation of intracortical and corticothalamic models of the alpha rhythm

Dear Dr Bastiaens,

I am pleased to inform you that your manuscript has been formally accepted for publication in PLOS Computational Biology. Your manuscript is now with our production department and you will be notified of the publication date in due course.

With kind regards,

Anita Estes
